# LEC: Linear Expectation Constraints for Selection-Conditioned Risk Control in Selective Prediction and Routing Systems

**Zhiyuan Wang** [1]   **Aniri** [2 3]   **Tianlong Chen** [4]   **Yue Zhang** [5]   **Heng Tao Shen** [6]   **Xiaoshuang Shi** [1]   **Kaidi Xu** [7]

## Abstract

Foundation models often generate unreliable answers, while heuristic uncertainty estimators fail to fully distinguish correct from incorrect outputs, causing users to accept erroneous answers without any statistical guarantee. We address this problem through selection-conditioned risk control, aiming to ensure that an accepted prediction has an error probability no larger than a user-specified risk level. To this end, we propose LEC, a principled framework that reframes selective prediction as a decision problem governed by a *linear expectation constraint* over selection and error indicators. This formulation directly controls the ratio between the expected number of accepted errors and the expected number of accepted predictions, which corresponds to the marginal error probability conditioned on selection. Under exchangeability, we derive a *finite-sample sufficient condition* that relies only on a held-out calibration set, enabling the computation of a risk-constrained, retention-maximizing threshold. Furthermore, we extend LEC to two-model routing systems: if the primary model's uncertainty exceeds its calibrated threshold, the input is delegated to a subsequent model, while maintaining system-level selection-conditioned error control. Experiments on both closed-ended and open-ended question answering (QA) and vision question answering (VQA) demonstrate that LEC maintains the prescribed risk level in accepted predictions and substantially improves sample retention compared to baselines.

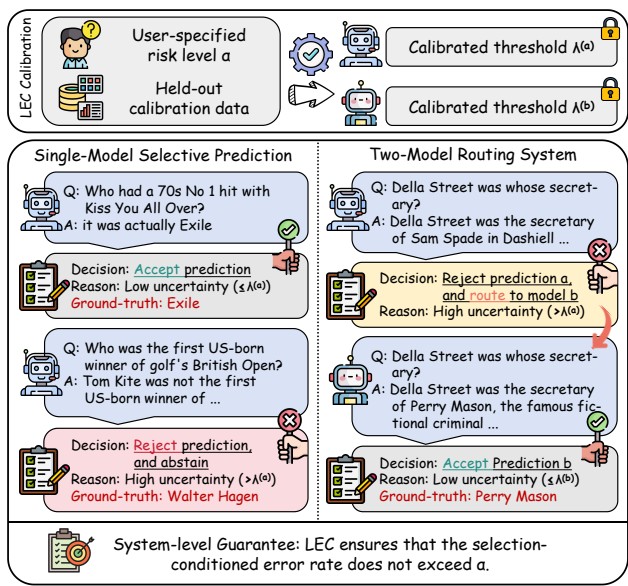

*Figure 1.* Illustration of selective prediction in single-model and two-model routing systems. By calibrating when to accept, escalate, or abstain, LEC provides system-level selection-conditioned error control. Code is available here.

[1]University of Electronic Science and Technology of China [2]LMU Munich [3]Munich Center for Machine Learning [4]University of North Carolina at Chapel Hill [5]Shandong University [6]Tongji University [7]City University of Hong Kong. Correspondence to: Xiaoshuang Shi <xsshi2013@gmail.com>, Kaidi Xu <kaidixu@cityu.edu.hk>.

*Proceedings of the 43rd International Conference on Machine Learning*, Seoul, South Korea. PMLR 306, 2026. Copyright 2026 by the author(s).

## 1. Introduction

Foundation models, like large language models (LLMs) and large vision-language models (LVLMs), are increasingly being integrated into real-world decision-making pipelines (Xiaolan et al., 2025; Brady et al., 2025; Singhal et al., 2025), where it is crucial to evaluate the reliability of their outputs and determine whether to trust them. Uncertainty quantification (UQ) is a promising approach to estimate the uncertainty of model predictions, with the uncertainty score serving as an indicator of whether the model's output is likely to be incorrect (Zhang et al., 2024; Wang et al., 2025c; Duan et al., 2024; 2025). In practice, when the model shows high uncertainty, its predictions should be clarified or abstained from to prevent the propagation of incorrect information.

However, when the model generates hallucinations or exhibits overconfidence in its erroneous predictions (Shorinwa et al., 2025; Atf et al., 2025), uncertainty scores derived from model logits or self-consistency measures may remain

low, leading users to accept incorrect answers without task-specific risk guarantees (Angelopoulos et al., 2024). Split conformal prediction (SCP) can transform heuristic uncertainty notions to statistically calibrated decision rules (Angelopoulos & Bates, 2021; Campos et al., 2024a; Tan et al., 2025). Assuming data exchangeability, SCP produces prediction sets that include ground-truth answers with at least a user-defined probability (Li et al., 2026). Nonetheless, set-valued predictions often contain unreliable candidates, leading to biased decision-making in downstream tasks (Wang et al., 2025a; Cresswell et al., 2025). In this paper, we investigate point prediction with provable finite-sample guarantees on the error rate among accepted predictions.

Although uncertainty scores cannot perfectly separate correct from incorrect predictions, selective prediction allows us to enforce a prespecified risk level (e.g., $\alpha$): a prediction is accepted if and only if its associated uncertainty score falls below a calibrated threshold, ensuring that the selection-conditioned error rate does not exceed $\alpha$. To achieve this in a principled way, we introduce LEC, which reframes selective prediction not as an uncertainty-ranking problem, but as a decision problem governed by a statistical constraint. The central idea is to express selection-conditioned error control as a constraint on the expectation of a linear functional involving two binary indicators: one capturing whether a prediction is selected and the other indicating whether it is incorrect. This formulation enables us to establish a finite-sample sufficient condition utilizing calibration uncertainty scores and error labels that, if satisfied, guarantees selection-conditioned error control for unseen test samples. Since this condition depends only on the empirical quantities from the calibration set, it yields a calibrated threshold that maximizes retention subject to the prescribed risk constraint.

We further extend LEC to a two-model routing framework. For each input, the system accepts the current model's prediction if its uncertainty falls below a calibrated threshold; otherwise, the input is routed to the subsequent model. If neither model satisfies its acceptance criterion, the system abstains. To preserve the statistical guarantee, we impose a linear expectation constraint on the system-level selection and error indicators, which enables joint calibration of model-specific thresholds with unified system-level selection-conditioned error control. Figure 1 illustrates examples of selective prediction in single-model prediction and two-model routing systems, where uncertainty serves as the decision signal for *accepting*, *routing*, or *abstaining*.

We evaluate LEC on four benchmarks across closed-ended and open-ended generation scenarios. In selective prediction of both single-model and two-model routing systems, LEC keeps the empirical accepted error rate below the prescribed risk level across feasible risk levels, consistent with its finite-sample selection-conditioned error guarantee. Compared to

confidence interval-based methods (Wang et al., 2026; Jung et al., 2025), LEC establishes tighter risk control while accepting more admissible samples (e.g., +9% on TriviaQA). Furthermore, across different UQ methods, admission functions, calibration-test split ratios, and sampling sizes under black-box scenarios, LEC maintains statistical rigor while consistently achieving higher power than the best baseline. These results highlight the practical effectiveness and generality of LEC, motivating its potential integration into real-world uncertainty-aware agentic systems.

## 2. Related Work

**SCP in LLMs.** SCP provides statistical guarantees of coverage for correct answers (Campos et al., 2024b). It evaluates the nonconformity (or residual) between model prediction and ground-truth on a calibration set, and then computes a rigorously calibrated threshold, which is applied to construct prediction/conformal sets at test time. Under exchangeability (Angelopoulos et al., 2023), these sets contain admissible answers with at least a user-specified probability. However, previous research predominantly focuses on *set-valued predictions* (Quach et al., 2024; Kaur et al., 2024; Wang et al., 2024b; 2025b;a; Li et al., 2026), which are not inherently actionable due to unreliable candidates, and can cause disparate impact (Cresswell et al., 2024; 2025). Our work targets selection-conditioned risk control over accepted *point predictions*, rather than conformal coverage.

**Risk Control in Selective Prediction.** Several frameworks grounded in significance testing (Jin & Candès, 2023; 2025) and confidence intervals (Bates et al., 2021) have been introduced to provide statistical error control for selective prediction (Jia et al., 2026). For instance, conformal alignment (Gui et al., 2024) and labeling (Huang et al., 2025) calculate conformal p-values and control false discoveries under multiple-testing formulations. To retain more admissible answers and accelerate test-time inference, COIN (Wang et al., 2026) constructs an upper confidence bound (UCB) for the system risk on calibration examples and computes a rigorous threshold for test-time selection, achieving PAC-style risk control (Park et al., 2020). Furthermore, Trust of Escalate (Jung et al., 2025) guarantees human agreement of cascaded LLM judges through Clopper-Pearson-style UCB (UCB-CLP) computation (Clopper & Pearson, 1934). While these methods provide valid risk control through high-probability upper confidence bounds, they are often overly conservative because they enforce worst-case tail control over the empirical risk estimate. In contrast, LEC directly constrains the expectation of a linear functional of selection and error indicators, yielding tighter yet still statistically valid selection-conditioned risk control.

# 3. Methodology

## 3.1. Notations and Problem Formulation

*1) Single-Model Selective Prediction with Selection-Conditioned Error Control.* Let $\mathcal{G}^{(a)} : \mathcal{X} \to \mathcal{Y}$ denote a pretrained model that maps an input prompt to a textual output. For a given prompt $x \in \mathcal{X}$ with an unknown ground-truth answer $y^* \in \mathcal{Y}$, the model produces a prediction $\hat{y}^{(a)} = \mathcal{G}^{(a)}(x) \in \mathcal{Y}$. We quantify the model's uncertainty for $x$ as $u^{(a)} = \mathcal{U}(x; \mathcal{G}^{(a)})$, where $\mathcal{U}(\cdot)$ denotes a scalar uncertainty function. Intuitively, small $u^{(a)}$ indicates high trustworthiness in $\hat{y}^{(a)}$. For a specified threshold $\lambda^{(a)}$, the prediction $\hat{y}^{(a)}$ is deemed admissible and accepted if $u^{(a)} \leq \lambda^{(a)}$. Let the admission function be

$$A(y^*, y) = \begin{cases} 1, & \text{if } y \in \mathcal{Y} \text{ is aligned with } y^*, \\ 0, & \text{otherwise.} \end{cases}$$

However, prior uncertainty methods are inherently imperfect and cannot fully separate correct from incorrect outputs (Liu et al., 2025). Thus, applying a fixed $\lambda^{(a)}$ at test time may admit some erroneous predictions. To mitigate this issue, our goal is to derive a statistically rigorous threshold $\hat{\lambda}^{(a)}$ that ensures the conditional probability that an accepted prediction is incorrect does not exceed a target risk level $\alpha$.

Formally, we define the selection indicator as $S^{(a)}\left(\lambda^{(a)}\right) = \mathbf{1}\left\{u^{(a)} \leq \lambda^{(a)}\right\}$, and the corresponding error indicator as $err^{(a)} = \mathbf{1}\left\{A(y^*, \hat{y}^{(a)}) = 0\right\}$. Our objective is to calibrate a statistically valid threshold $\hat{\lambda}^{(a)}$ such that

$$\Pr\left(err^{(a)} = 1 \mid S^{(a)}(\hat{\lambda}^{(a)}) = 1\right) \leq \alpha, \quad \alpha \in (0, 1). \tag{1}$$

We refer to the left-hand side of Eq. (1) as the *selection-conditioned error rate*: the marginal error probability of a prediction after it has been selected by the calibrated rule.

*2) Two-Model Routing with System-Level Selection-Conditioned Error Control.* Under a specific uncertainty function $\mathcal{U}(\cdot)$, the uncertainty scores of model $\mathcal{G}^{(a)}$ on test examples may cluster too tightly in a low range, making it impossible to achieve small target risk levels. Moreover, when $\mathcal{G}^{(a)}$ has limited predictive ability, many challenging or critical prompts may be abstained from, leading to reduced system efficiency. To alleviate these issues, we develop a collaborative routing mechanism that dynamically delegates uncertain samples to another model with stronger accuracy or a more discriminative uncertainty profile, while controlling the system-level selection-conditioned error rate.

Formally, we define the alternative model as $\mathcal{G}^{(b)} : \mathcal{X} \to \mathcal{Y}$. For a given prompt $x$, when the estimated uncertainty $u^{(a)}$ exceeds $\lambda^{(a)}$, we route the prompt to $\mathcal{G}^{(b)}$. We denote the prediction of $\mathcal{G}^{(b)}$ as $\hat{y}^{(b)} \in \mathcal{Y}$, along with the corresponding uncertainty $u^{(b)} = \mathcal{U}(x; \mathcal{G}^{(b)})$. Similarly, if $u^{(b)}$ does not exceed the threshold $\lambda^{(b)}$ of model $\mathcal{G}^{(b)}$, we trust $\hat{y}^{(b)}$; otherwise, the two-model routing system abstains from the prompt $x$. We define the selection indicator of model $\mathcal{G}^{(b)}$ as

$$S^{(b)}\left(\lambda^{(a)}, \lambda^{(b)}\right) = \mathbf{1}\left\{u^{(a)} > \lambda^{(a)} \wedge u^{(b)} \leq \lambda^{(b)}\right\},$$

and the error indicator as $err^{(b)} = \mathbf{1}\left\{A(y^*, \hat{y}^{(b)}) = 0\right\}$.

The two-model routing system $\mathcal{G}$ integrates $\mathcal{G}^{(a)}$ and $\mathcal{G}^{(b)}$, with the system-level selection indicator

$$S\left(\lambda^{(a)}, \lambda^{(b)}\right) = S^{(a)}\left(\lambda^{(a)}\right) + S^{(b)}\left(\lambda^{(a)}, \lambda^{(b)}\right) \in \{0, 1\}.$$

The system-level accepted-error indicator is

$$err = \mathbf{1}\{S^{(a)}(\lambda^{(a)}) = 1 \wedge err^{(a)} = 1\} \\ + \mathbf{1}\{S^{(b)}(\lambda^{(a)}, \lambda^{(b)}) = 1 \wedge err^{(b)} = 1\}.$$

When $S(\lambda^{(a)}, \lambda^{(b)}) = 1$, the prediction from either $\mathcal{G}^{(a)}$ or $\mathcal{G}^{(b)}$ is accepted. We aim to jointly calibrate $(\lambda^{(a)}, \lambda^{(b)})$ and obtain statistically rigorous thresholds $(\hat{\lambda}^{(a)}, \hat{\lambda}^{(b)})$ such that

$$\Pr\left(err = 1 \mid S\left(\hat{\lambda}^{(a)}, \hat{\lambda}^{(b)}\right) = 1\right) \leq \alpha, \quad \alpha \in (0, 1). \tag{2}$$

This guarantees that the overall two-model routing system performs selective prediction with system-level selection-conditioned error control.

## 3.2. Threshold Calibration for Single-Model Settings

We begin by describing how to calibrate a statistically valid threshold $\hat{\lambda}^{(a)}$ for $\mathcal{G}^{(a)}$. Following the standard split calibration protocol (Papadopoulos et al., 2002), the dataset is partitioned into a calibration set and a test set. The threshold is learned solely from the calibration data for a user-specified risk level $\alpha$, and is then fixed during test-time evaluation.

**From selection-conditioned error control to linear expectation constraint.** For a fixed threshold $\lambda^{(a)}$, recall the selection and error indicators $S^{(a)}(\lambda^{(a)})$ and $err^{(a)}$. We further define the joint indicator as $Z^{(a)}(\lambda^{(a)}) = S^{(a)}(\lambda^{(a)}) \cdot err^{(a)}$, which equals 1 if and only if we accept the prediction and the model errs. The selection-conditioned error rate can then be written as

$$\text{SCER}^{(a)}(\lambda^{(a)}) = \Pr\left(err^{(a)} = 1 \mid S^{(a)}(\lambda^{(a)}) = 1\right) \\ = \frac{\Pr\left(err^{(a)} = 1 \wedge S^{(a)}(\lambda^{(a)}) = 1\right)}{\Pr\left(S^{(a)}(\lambda^{(a)}) = 1\right)} = \frac{\mathbb{E}[Z^{(a)}(\lambda^{(a)})]}{\mathbb{E}[S^{(a)}(\lambda^{(a)})]}. \tag{3}$$

As long as $\mathbb{E}[S^{(a)}(\lambda^{(a)})] > 0$, $\text{SCER}^{(a)}(\lambda^{(a)}) \leq \alpha$ is equivalent to a constraint on the expectation of a linear functional of the selection and error indicators:

$$\mathbb{E}\left[Z^{(a)}(\lambda^{(a)}) - \alpha S^{(a)}(\lambda^{(a)})\right] \leq 0. \tag{4}$$

Intuitively, the random variable $Z^{(a)} - \alpha S^{(a)}$ measures *accepted error count minus $\alpha$ times selection count* on a single example; if its expectation is non-positive, then the marginal error probability conditioned on selection does not exceed $\alpha$.

**Finite-sample sufficient condition.** To enforce the population constraint in Eq. (4) using only the calibration data, we derive a finite-sample sufficient condition. Let the calibration set be $\mathcal{D}_{\text{cal}} = \{(u_i^{(a)}, err_i^{(a)})\}_{i=1}^n$, with $\{S_i^{(a)}\}_{i=1}^n$, and let $u_{(1)}^{(a)} \leq \cdots \leq u_{(n)}^{(a)}$ denote the calibration uncertainty scores sorted in ascending order, with corresponding error indicators $err_{(j)}^{(a)}$. For a candidate threshold $\lambda^{(a)}$, we define

$$k^{(a)}(\lambda^{(a)}) = \#\{i : S_i^{(a)}(\lambda^{(a)}) = 1\} = \#\{i : u_i^{(a)} \leq \lambda^{(a)}\}$$

as the number of calibration data points that would be accepted at threshold $\lambda^{(a)}$. Motivated by the standard leave-one-out correction in distribution-free calibration, we use the following finite-sample sufficient condition, whose validity under exchangeability is established in Appendix A.1:

$$\sum_{j=1}^{k^{(a)}(\lambda^{(a)})} \left(err_{(j)}^{(a)} - \alpha\right) \leq -1. \quad (5)$$

We then define the feasible set of thresholds at level $\alpha$ as

$$\Lambda_\alpha^{(a)} = \left\{\lambda^{(a)} : \sum_{j=1}^{k^{(a)}(\lambda^{(a)})} \left(err_{(j)}^{(a)} - \alpha\right) \leq -1\right\}. \quad (6)$$

**Calibrated retention-maximizing threshold.** Among all thresholds in $\Lambda_\alpha^{(a)}$, we choose the largest feasible one to maximize the acceptance rate:

$$\begin{aligned} \hat{\lambda}^{(a)} &= \sup \Lambda_\alpha^{(a)} \\ &= \sup \left\{\lambda^{(a)} : \sum_{j=1}^{k^{(a)}(\lambda^{(a)})} \left(err_{(j)}^{(a)} - \alpha\right) \leq -1\right\}. \end{aligned} \quad (7)$$

If $\Lambda_\alpha^{(a)}$ is empty, we declare the target risk level $\alpha$ infeasible for $\mathcal{G}^{(a)}$ and abstain from all samples at this level.

**Theorem 3.1** (Single-model selection-conditioned error control)**.** *Assume that calibration and test examples are exchangeable (Angelopoulos et al., 2023). Let $\hat{\lambda}^{(a)}$ be defined by Eq. (7) using $\mathcal{D}_{\text{cal}}$. Then, for a new test sample $(x_{n+1}, y_{n+1}^*)$ with $(u_{n+1}^{(a)}, err_{n+1}^{(a)})$,*

$$\Pr\left(err_{n+1}^{(a)} = 1 \mid u_{n+1}^{(a)} \leq \hat{\lambda}^{(a)}\right) \leq \alpha,$$

*where the probability is taken over the joint randomness of the calibration set and the test sample (marginal guarantee). If $\Pr(u_{n+1}^{(a)} \leq \hat{\lambda}^{(a)}) = 0$, the guarantee is vacuous.*

A complete proof of Theorem 3.1 is given in Appendix A.1. At test time, for a new instruction $x_{n+1}$, we obtain the model prediction $\hat{y}_{n+1}^{(a)}$ with uncertainty $u_{n+1}^{(a)}$. We accept $\hat{y}_{n+1}^{(a)}$ if and only if $u_{n+1}^{(a)} \leq \hat{\lambda}^{(a)}$; otherwise, we abstain.

### 3.3. Threshold Calibration for Two-Model Routing

We now extend the above calibration procedure to the two-model routing system $\mathcal{G}$. For each example $i$, we observe uncertainties $(u_i^{(a)}, u_i^{(b)})$ and error indicators $(err_i^{(a)}, err_i^{(b)})$. Given thresholds $(\lambda^{(a)}, \lambda^{(b)})$, routing is defined by the selection indicators $S_i^{(a)}(\lambda^{(a)})$ and $S_i^{(b)}(\lambda^{(a)}, \lambda^{(b)})$. The system-level selection indicator is

$$S_i(\lambda^{(a)}, \lambda^{(b)}) = S_i^{(a)}(\lambda^{(a)}) + S_i^{(b)}(\lambda^{(a)}, \lambda^{(b)}) \in \{0, 1\}.$$

The accepted-error indicator is

$$\begin{aligned} err_i = &\mathbf{1}\{S_i^{(a)}(\lambda^{(a)}) = 1 \wedge err_i^{(a)} = 1\} + \\ &\mathbf{1}\{S_i^{(b)}(\lambda^{(a)}, \lambda^{(b)}) = 1 \wedge err_i^{(b)} = 1\}, \end{aligned}$$

which remains binary because routing selects at most one prediction. We also define the system-level joint indicator as

$$Z_i(\lambda^{(a)}, \lambda^{(b)}) = S_i^{(a)}(\lambda^{(a)}) \cdot err_i^{(a)} + S_i^{(b)}(\lambda^{(a)}, \lambda^{(b)}) \cdot err_i^{(b)}.$$

**From system-level selection-conditioned error control to expectation constraint.** The system-level selection-conditioned error rate at thresholds $(\lambda^{(a)}, \lambda^{(b)})$ is

$$\text{SCER}(\lambda^{(a)}, \lambda^{(b)}) = \frac{\mathbb{E}[Z(\lambda^{(a)}, \lambda^{(b)})]}{\mathbb{E}[S(\lambda^{(a)}, \lambda^{(b)})]}.$$

Whenever $\mathbb{E}[S(\lambda^{(a)}, \lambda^{(b)})] > 0$, $\text{SCER}(\lambda^{(a)}, \lambda^{(b)}) \leq \alpha$ is equivalent to a linear expectation inequality

$$\mathbb{E}\left[Z(\lambda^{(a)}, \lambda^{(b)}) - \alpha S(\lambda^{(a)}, \lambda^{(b)})\right] \leq 0. \quad (8)$$

This condition generalizes the single-model constraint to the routing system and captures the difference between the system-level accepted-error count and the $\alpha$-fraction of accepted samples.

**Finite-sample sufficient condition.** To enforce Eq. (8) from calibration points, we construct an empirical sufficient condition. Let $\mathcal{D}_{\text{cal}}^{\text{sys}} = \{(u_i^{(a)}, u_i^{(b)}, err_i^{(a)}, err_i^{(b)})\}_{i=1}^n$ denote the calibration set for the two-model routing system. Using the same leave-one-out correction for the system-level pair $(Z_i, S_i)$, we obtain the following finite-sample sufficient condition, with the validity argument deferred to Appendix A.2:

$$\sum_{i=1}^n \left(Z_i(\lambda^{(a)}, \lambda^{(b)}) - \alpha S_i(\lambda^{(a)}, \lambda^{(b)})\right) \leq -1. \quad (9)$$

We then obtain the feasible set of two-model threshold pairs

$$\begin{aligned} \Lambda_\alpha^{(a,b)} = \Big\{(\lambda^{(a)}, \lambda^{(b)}) : \sum_{i=1}^n \Big(Z_i(\lambda^{(a)}, \lambda^{(b)}) \\ - \alpha S_i(\lambda^{(a)}, \lambda^{(b)})\Big) \leq -1\Big\}. \end{aligned} \quad (10)$$

**Calibrated retention-maximizing thresholds.** Among all pairs $(\lambda^{(a)}, \lambda^{(b)}) \in \Lambda_\alpha^{(a,b)}$, we choose those that maximize the empirical acceptance rate of the routing system:

$$(\hat{\lambda}^{(a)}, \hat{\lambda}^{(b)}) = \operatorname*{argmax}_{(\lambda^{(a)}, \lambda^{(b)}) \in \Lambda_\alpha^{(a,b)}} \frac{1}{n} \sum_{i=1}^n S_i(\lambda^{(a)}, \lambda^{(b)}). \quad (11)$$

If $\Lambda_\alpha^{(a,b)}$ is empty, the risk level $\alpha$ is infeasible for the two-model routing system, and the system abstains on all inputs.

**Theorem 3.2** (Selection-conditioned error control for the two-model routing system). *Assume calibration and test examples are exchangeable. Let $(\hat{\lambda}^{(a)}, \hat{\lambda}^{(b)})$ be any solution of Eq.* (11)*. Then the two-model routing system satisfies*

$$\Pr\left(err_{n+1} = 1 \mid S_{n+1}(\hat{\lambda}^{(a)}, \hat{\lambda}^{(b)}) = 1\right) \leq \alpha,$$

*where the probability is taken over the joint randomness of calibration and test samples (marginal guarantee). If* $\Pr(S_{n+1}(\hat{\lambda}^{(a)}, \hat{\lambda}^{(b)}) = 1) = 0$*, the guarantee is vacuous.*

See a proof of Theorem 3.2 in Appendix A.2. At test time, each user instruction $x_{n+1}$ is processed as follows: we accept $\hat{y}_{n+1}^{(a)}$ via $\mathcal{G}^{(a)}$ if $u_{n+1}^{(a)} \leq \hat{\lambda}^{(a)}$; otherwise we route the prompt to $\mathcal{G}^{(b)}$ and accept $\hat{y}_{n+1}^{(b)}$ if $u_{n+1}^{(b)} \leq \hat{\lambda}^{(b)}$. If neither condition is satisfied, the system abstains. Our analysis highlights that system-level selection-conditioned error control is preserved as long as the routing policy is deterministic and each example is routed to at most one model. The statistical guarantees arise from the linear decomposition, rather than any model-specific assumptions.

The above two-model calibration can readily be extended to routing systems with more than two models. In Appendix B, we outline how LEC extends to general multi-model routing systems, offering a principled mechanism for unified selection-conditioned error control across routing policies of arbitrary depth.

# 4. Experiments

## 4.1. Experimental Settings

**Benchmarks and Models.** (1) QA: We evaluate LEC on the CommonsenseQA (closed-ended) (Talmor et al., 2019) and TriviaQA (open-ended) (Joshi et al., 2017) datasets using eight LLMs, including LLaMA (Touvron et al., 2023), Qwen (Bai et al., 2023), Vicuna (Zheng et al., 2023), and OpenChat (Wang et al., 2024a) families. (2) VQA: We also consider the ScienceQA (closed-ended) (Lu et al., 2022) and MM-Vet v2 (open-ended) (Yu et al., 2024) benchmarks, using four LVLMs, including LLaVA1.5 (Liu et al., 2023), LLaVA-NeXT (Liu et al., 2024), and InternVL2 (Chen et al., 2024) groups. We omit suffixes such as "hf" and "Instruct".

**Evaluation Metrics.** Following previous evaluation protocols (Jung et al., 2025; Wang et al., 2026), we evaluate

the statistical validity of LEC by verifying that the empirical error rate among accepted predictions does not exceed the target risk level. Our theoretical target is the selection-conditioned error rate,

$$\text{SCER}(\lambda) = \Pr(err = 1 \mid S(\lambda) = 1) = \frac{\mathbb{E}[S(\lambda) \cdot err]}{\mathbb{E}[S(\lambda)]}.$$

In the experimental figures, the y-axis label "FDR" is retained as a compact plotting shorthand for the observed false-discovery proportion among accepted predictions:

$$\widehat{\text{FDP}}_{\text{acc}} = \frac{\sum_{i \in \mathcal{D}_{\text{test}}} S_i(\hat{\lambda})\, err_i}{\left(\sum_{i \in \mathcal{D}_{\text{test}}} S_i(\hat{\lambda})\right) \vee 1}.$$

This empirical quantity is utilized only to summarize test-time performance and serves as a plug-in estimate of the selection-conditioned error rate controlled by our theory. Throughout the paper, all theoretical guarantees are stated in terms of selection-conditioned error control. We further assess power, defined as the proportion of aligned test predictions accepted by the method among all aligned predictions. In two-model routing, we additionally report the allocation ratio of accepted samples across the two models.

**Baselines.** *1) Single-model:* We consider UCB-based methods that control the accepted error rate by computing UCBs on the system risk from calibration data. Specifically, we implement two variants: `UCB-HFD`, which derives the UCB using Hoeffding's inequality (Hoeffding, 1963), and `UCB-CLP`, which adopts the exact Clopper–Pearson interval. These two variants abstract the core confidence-bound-based risk control mechanism used in prior single-model methods such as COIN (Wang et al., 2026). *2) Two-model routing:* We extend the UCB-based approach to the routing setting by applying the same confidence-bound-based risk control to the system-level selection and error indicators. We consider `UCB-CLP-Routing`, corresponding to the cascaded judge in Jung et al. (2025), as well as `UCB-HFD-Routing`, which replaces the Clopper-Pearson bound with Hoeffding's inequality for a distribution-free variant. We do not consider routing with more than two models, as it only increases the number of threshold parameters and leads to nested threshold searches during calibration, without altering the formulation or its statistical guarantees.

**Alignment Criteria.** We use sentence similarity (Reimers & Gurevych, 2019b) with a 0.6 threshold to decide whether the model's answer is aligned with the ground truth in the admission function $A$ by default. We also use bi-entailment (Kuhn et al., 2023) and LLM-as-a-Judge (Zhang et al., 2024).

**Uncertainty Estimator $\mathcal{U}$.** In closed-ended QA and VQA, we estimate uncertainty scores by computing the predictive entropy (PE) (Kadavath et al., 2022). We use the softmax output of model logits by default. We also generate multiple

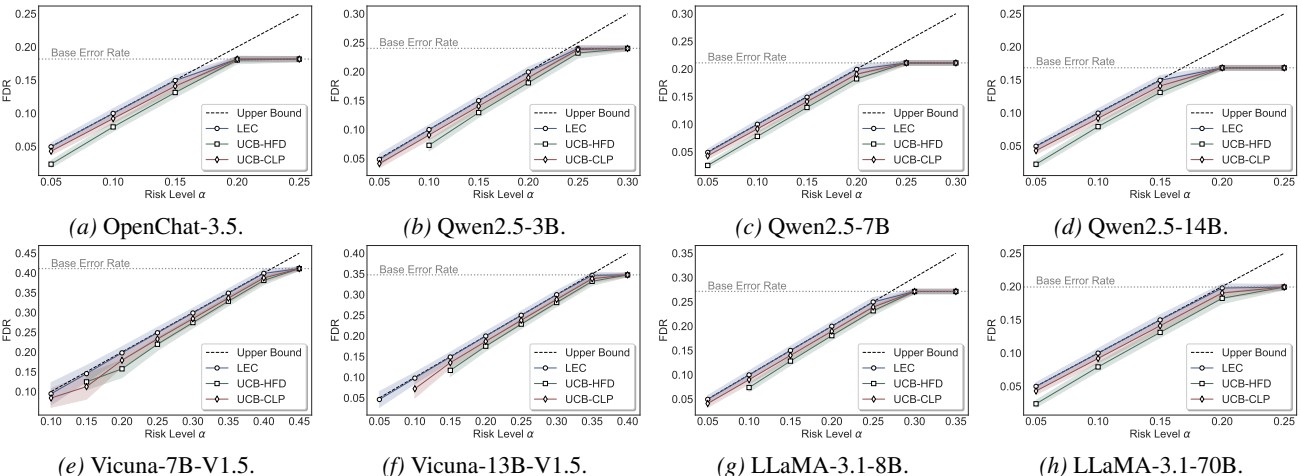

*Figure 2.* Test-time empirical selection-conditioned error rate on the CommonsenseQA dataset (mean±std). The y-axis label "FDR" denotes the observed fraction of erroneous predictions among accepted predictions. LEC provides tighter risk control while maintaining the prescribed risk level.

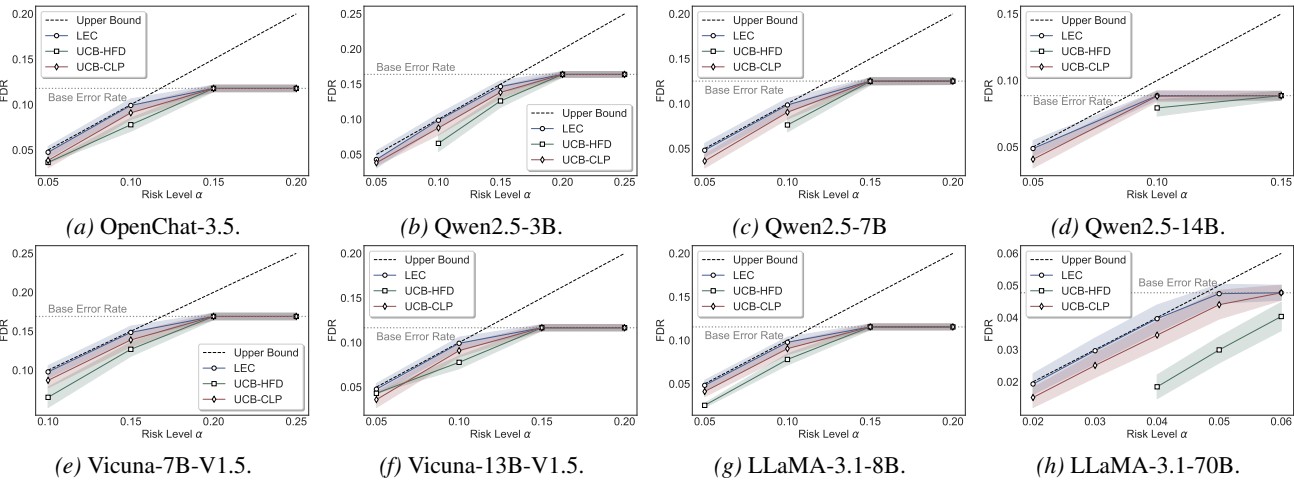

*Figure 3.* Test-time empirical selection-conditioned error rate on the TriviaQA dataset (mean±std). LEC provides tighter risk control while maintaining the prescribed risk level.

answers per input and employ sampling frequency as the generative probability (Wang et al., 2025d). In open-ended QA and VQA, we compute the black-box semantic entropy (SE) (Farquhar et al., 2024) by default. Moreover, we use the sum of eigenvalues of the graph laplacian (EigV), degree matrix (Deg), and eccentricity (Ecc) (Lin et al., 2024). We also consider the length-normalized PE (Malinin & Gales, 2021) of the model's output itself (SELF).

**Hyperparameters.** Following previous work (Wang et al., 2026), we employ beam search (num_beams=5) to obtain the most likely generation as the model output. By default, for open-domain QA, we sample 10 answers per input for UQ. In addition, we fix the calibration-test split ratio to 0.5.

We provide the details of additional experimental settings in Appendix C. Following prior research (Quach et al., 2024),

we randomly split the calibration and test samples 500 times and report the mean and standard deviation (mean±std). We annotate this information alongside the subsequent results.

## 4.2. Evaluations in Single-Model Selective Prediction

**Statistical Validity.** We first evaluate LEC in single-model selective prediction settings. As demonstrated in Figures 2 and 3, across both CommonsenseQA and TriviaQA datasets and eight LLMs, LEC consistently matches the prescribed selection-conditioned risk target: the empirical accepted error rate, averaged over 500 random splits, remains below the target risk level. For example, on CommonsenseQA with a risk level of 0.05, LEC achieves an average empirical accepted error rate of 0.0497 when applied to OpenChat-3.5.

**Tighter Risk Control.** Beyond statistical validity, we exam-

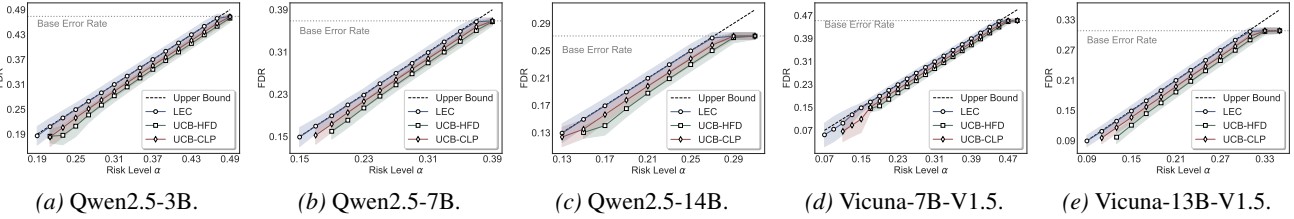

*Figure 4.* Test-time empirical selection-conditioned error rate on TriviaQA with entailment for correctness evaluation (mean±std).

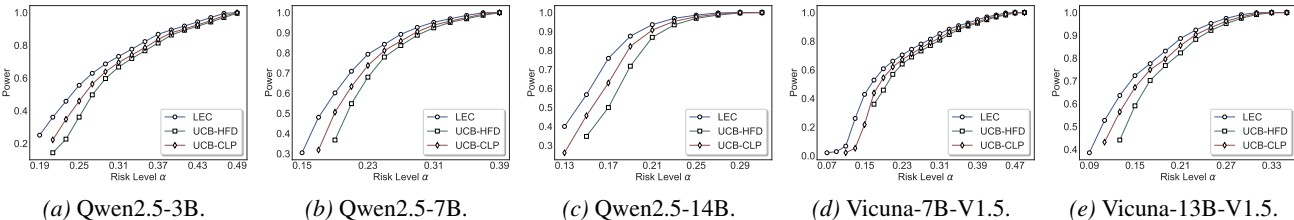

*Figure 5.* Test-time Power on the TriviaQA dataset with entailment for correctness evaluation (mean).

*Table 1.* Power comparison on the TriviaQA dataset (mean).

| LLMs | Methods / $\alpha$ | 0.05 | 0.1 | 0.15 | 0.2 | 0.25 |
|---|---|---|---|---|---|---|
| OpenChat-3.5 | UCB-CLP | 0.6684 | 0.9294 | 1.0 | 1.0 | 1.0 |
| | UCB-HFD | 0.6091 | 0.8884 | 1.0 | 1.0 | 1.0 |
| | LEC | **0.7230** | **0.9521** | **1.0** | **1.0** | **1.0** |
| Qwen2.5-3B | UCB-CLP | 0.2376 | 0.5219 | 0.8554 | 1.0 | 1.0 |
| | UCB-HFD | - | 0.3882 | 0.7772 | 1.0 | 1.0 |
| | LEC | **0.2706** | **0.5998** | **0.9081** | **1.0** | **1.0** |
| Qwen2.5-7B | UCB-CLP | 0.3905 | 0.8331 | 1.0 | 1.0 | 1.0 |
| | UCB-HFD | - | 0.7396 | 0.9990 | 1.0 | 1.0 |
| | LEC | **0.4889** | **0.8850** | **1.0** | **1.0** | **1.0** |
| Qwen2.5-14B | UCB-CLP | 0.6240 | 0.9987 | 1.0 | 1.0 | 1.0 |
| | UCB-HFD | - | 0.9718 | 1.0 | 1.0 | 1.0 |
| | LEC | **0.7193** | **1.0** | **1.0** | **1.0** | **1.0** |
| Vicuna-7B-V1.5 | UCB-CLP | - | 0.5686 | 0.8630 | 1.0 | 1.0 |
| | UCB-HFD | - | 0.4228 | 0.8068 | 0.9999(5) | 1.0 |
| | LEC | - | **0.6508** | **0.9208** | **1.0** | **1.0** |
| Vicuna-13B-V1.5 | UCB-CLP | 0.5602 | 0.8944 | 1.0 | 1.0 | 1.0 |
| | UCB-HFD | 0.5241 | 0.8364 | 1.0 | 1.0 | 1.0 |
| | LEC | **0.6545** | **0.9342** | **1.0** | **1.0** | **1.0** |
| LLaMA-3.1-8B | UCB-CLP | 0.7143 | 0.9396 | 1.0 | 1.0 | 1.0 |
| | UCB-HFD | 0.5339 | 0.9039 | 1.0 | 1.0 | 1.0 |
| | LEC | **0.7538** | **0.9612** | **1.0** | **1.0** | **1.0** |
| LLaMA-3.1-70B | UCB-CLP | 0.9935 | 1.0 | 1.0 | 1.0 | 1.0 |
| | UCB-HFD | 0.9503 | 1.0 | 1.0 | 1.0 | 1.0 |
| | LEC | **0.9996** | **1.0** | **1.0** | **1.0** | **1.0** |

ine how tightly different methods control the system-level risk under the same target risk constraint. As presented in Figure 2 and Figure 3, across both datasets and all LLMs, `LEC` consistently operates near the target risk level, while UCB-based baselines, including those using exact Clopper-Pearson-style UCB, remain well below it, indicating more conservative behavior. For instance, on TriviaQA utilizing the Qwen2.5-3B model, `LEC` achieves an empirical accepted error rate of 0.0987, whereas `UCB-CLP` attains 0.0878, and `UCB-HFD` fails to identify feasible thresholds due to overly conservative UCB.

This tighter control allows `LEC` to retain substantially more

samples without compromising the user-specified risk constraint. As illustrated in Table 1, this difference in tightness is further reflected in the power of each method. Across all evaluated LLMs and risk levels, `LEC` consistently achieves higher power than UCB-based baselines, indicating that it admits more valid predictions under the same statistical constraint. In contrast, the conservative nature of UCB-based methods, particularly those based on Hoeffding's inequality, often leads to substantially reduced power, or even the absence of feasible thresholds at low risk levels. For example, at a risk level of 0.05 on TriviaQA with the Qwen2.5-14B model, `LEC` achieves a power of 0.7193, retaining 9.5% more admissible samples than `UCB-CLP`, while `UCB-HFD` fails to yield any feasible threshold at this level.

**Robustness Across Evaluation Settings.** We further examine whether the observed advantages of `LEC` are sensitive to specific evaluation settings. As shown in Figures 4 and 5, across five LLMs and all tested risk levels, `LEC` consistently maintains tight selection-conditioned error control and achieves higher power than UCB-based baselines under the same statistical constraints, with bi-entailment as the alignment criterion in function $A$.

### 4.3. Evaluations in Two-Model Routing Systems

We denote by `LEC-Routing` the routing strategy obtained by applying the proposed linear expectation constraint to the two-model routing setting, where joint thresholds are calibrated over the system-level selection and error indicators to ensure system-level selection-conditioned error control across the entire routing pipeline.

**System-Level Selection-Conditioned Risk Control.** We compare `LEC-Routing` with UCB-based routing baselines (Jung et al., 2025). In addition, we consider a naive

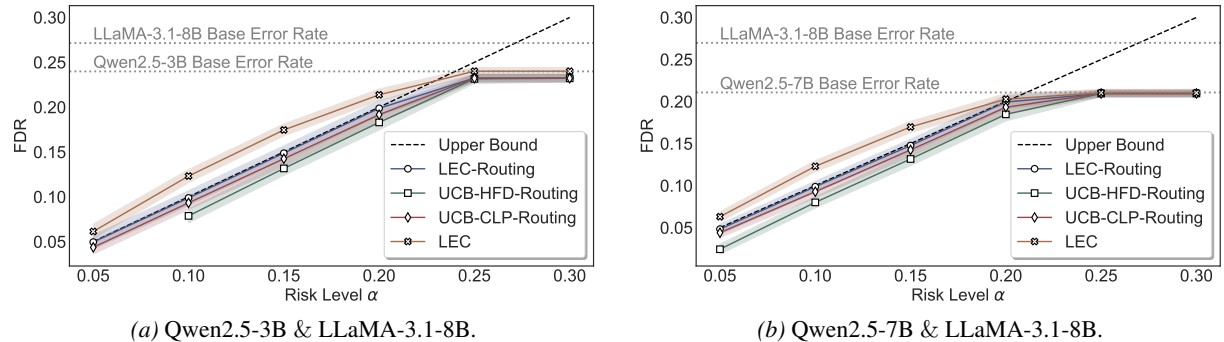

*Figure 6.* Test-time empirical system-level selection-conditioned error rate of two-model routing on CommonsenseQA (mean±std).

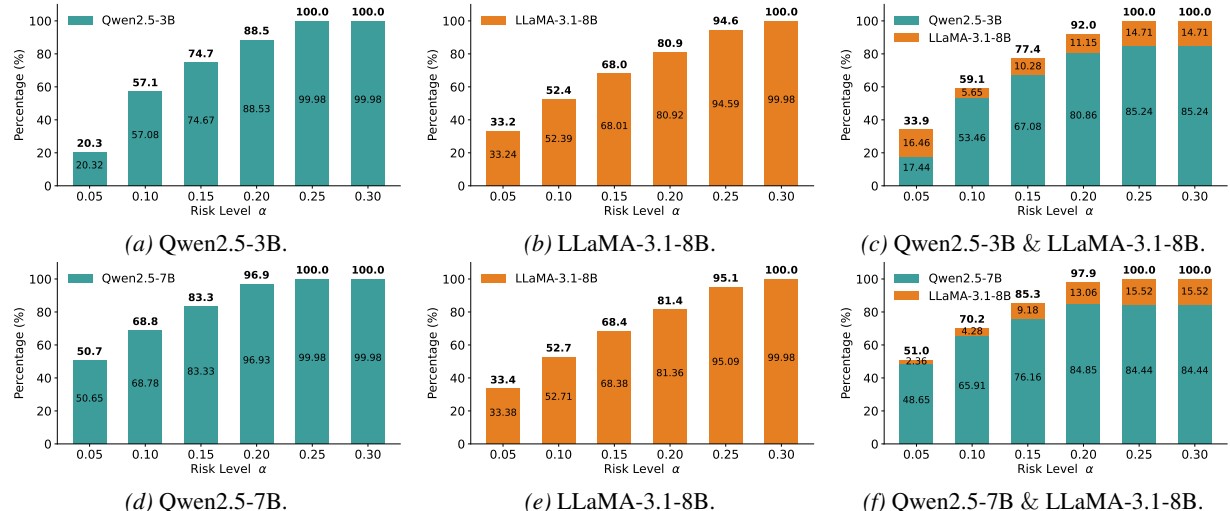

*Figure 7.* Allocation ratio of accepted test samples in two-model routing systems on the CommonsenseQA dataset (mean).

routing variant that calibrates thresholds for each model independently using LEC at the same target, without joint threshold calibration. Figure 6 shows that LEC-Routing consistently maintains valid and tight system-level selection-conditioned error control on CommonsenseQA by employing both Qwen2.5-3B and Qwen2.5-7B as primary models and selectively delegating inputs to the LLaMA-3.1-8B model, while UCB-HFD-Routing and UCB-CLP-Routing exhibit more conservative behavior. Notably, applying LEC without joint threshold calibration does not achieve valid system-level guarantees, which highlights the necessity of joint threshold calibration for achieving reliable system-level risk control in routing systems.

**Routing Allocation.** We further examine the allocation of accepted test samples under two-model routing. As shown in Figure 7, at different risk levels and model pairs, the distribution of accepted samples adapts to the risk budget and the uncertainty profiles of the models: the primary model handles a substantial portion of the accepted samples when its predictions are sufficiently reliable, while uncertain cases are selectively delegated to the secondary model. This adap-

tive allocation leads to both higher system-level coverage and improved cost-efficiency under the prescribed risk constraint. For example, at $\alpha = 0.05$, using Qwen2.5-3B alone accepts only $20.3\%$ of the test samples. In contrast, under LEC-Routing with Qwen2.5-3B as the primary model and LLaMA-3.1-8B as the secondary model, the system accepts $33.9\%$ of the samples in total, with $17.44\%$ handled by Qwen2.5-3B and an additional $16.46\%$ selectively routed to LLaMA-3.1-8B—representing a $13.6\%$ absolute increase in accepted samples over using the primary model alone.

Importantly, routing does not trivially favor the secondary model. At the same risk level, Qwen2.5-7B alone accepts $50.7\%$ of the samples, while LLaMA-3.1-8B alone accepts only $33.4\%$. In this case, LEC-Routing still achieves an acceptance rate of $51.0\%$, with the majority of samples ($48.65\%$) processed by the more efficient primary model Qwen2.5-7B. This demonstrates that LEC-Routing dynamically balances model usage based on reliability and risk, rather than indiscriminately escalating samples. Overall, these results highlight a principled trade-off between efficiency and cost: depending on the risk level and model

*Table 2.* Comparison of the number of accepted correct samples at test time on the CommonsenseQA dataset (mean).

| LLMs / $\alpha$ | 0.05 | 0.1 | 0.15 | 0.2 | 0.25 | 0.3 |
|---|---|---|---|---|---|---|
| Qwen2.5-3B | 965 | 2569 | 3174 | 3540 | 3797 | 3797 |
| LLaMA-3.1-8B | 1579 | 2357 | 2890 | 3238 | 3549 | 3643 |
| Qwen2.5-3B & LLaMA-3.1-8B | **1610** | **2663** | **3293** | **3686** | **3836** | **3836** |
| Qwen2.5-7B | 2392 | 3078 | 3523 | 3858 | 3924 | 3924 |
| LLaMA-3.1-8B | 1577 | 2360 | 2892 | 3237 | 3546 | 3629 |
| Qwen2.5-7B & LLaMA-3.1-8B | **2413** | **3144** | **3615** | **3896** | **3928** | **3928** |

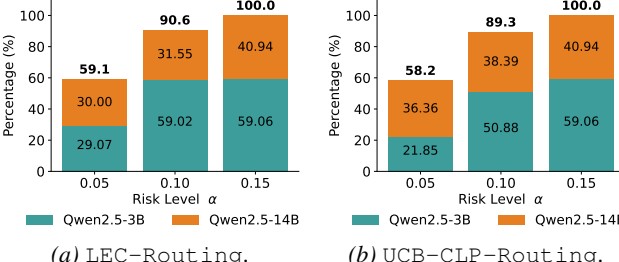

*(a)* `LEC-Routing`.     *(b)* `UCB-CLP-Routing`.

*Figure 8.* Comparison in the allocation ratio of accepted samples in two-model routing systems on the TriviaQA dataset (mean).

characteristics, `LEC-Routing` maximizes system-level utility by invoking the secondary models only when necessary, while preserving rigorous system-level selection-conditioned error guarantees.

Also, we show that `LEC-Routing` has the potential to increase the effective set of correct predictions under system-level selection-conditioned error control. Table 2 shows that two-model routing under `LEC-Routing` consistently retains more correct samples than using either model alone across all risk levels. For instance, at a risk level of $\alpha = 0.05$, routing Qwen2.5-3B with LLaMA-3.1-8B admits 1610 correct samples, compared to 965 and 1579 correct samples when using Qwen2.5-3B or LLaMA-3.1-8B alone.

Finally, we illustrate a comparison of the routing behavior between `LEC-Routing` and `UCB-CLP-Routing`. As demonstrated in Figure 8, under the same target risk levels, `LEC-Routing` tends to prioritize the primary model more effectively, while retaining a larger set of accepted samples overall. By contrast, `UCB-CLP-Routing` exhibits a more conservative allocation pattern, with a higher reliance on the secondary model. These results further suggest that the tighter system-level risk control of our `LEC` framework can translate into more efficient routing decisions, although the extent of this advantage may vary across settings.

Additional experimental results for both single-model and two-model routing systems are reported in Appendix D.

## 5. Conclusion

In this paper, we introduce `LEC`, a principled formulation that frames selective prediction as a decision problem gov-

erned by a linear expectation constraint over selection and error indicators. By directly constraining the expected system-level risk, `LEC` departs from conventional UCB-based approaches that rely on worst-case tail bounds, and instead characterizes a tighter feasible region for admissible decisions. We demonstrate that our framework applies naturally to single-model selective prediction and two-model routing systems, where joint calibration over system-level indicators is essential for reliable risk control. Empirically, `LEC` consistently keeps the observed accepted-error rate below the prescribed risk level across various closed-ended and open-ended QA and VQA datasets. In routing settings, risk-aware calibration enables adaptive allocation of samples across models and can, in favorable regimes, increase the number of accepted correct predictions relative to single-model deployment, while supporting principled trade-offs between coverage, accuracy, and cost. Overall, `LEC` provides a general foundation for system-level risk control in selective prediction and routing. Future work may explore tighter characterizations of when routing yields maximal benefits, extend the framework to broader task-specific risk measures, and integrate `LEC` with more expressive routing architectures to support reliable decision-making in complex, multi-agent systems.

## Acknowledgements

The paper was supported by Noncommunicable Chronic Diseases-National Science and Technology Major Project (2025ZD0551300, 2025ZD0551302).

## Impact Statement

`LEC` offers a foundation for integrating foundation models into high-stakes scenarios that require transparent and auditable reliability guarantees. Relying solely on held-out calibration data and exchangeability assumptions, the framework remains applicable in black-box settings and across heterogeneous data sources. We believe this work opens avenues for future research on composable risk control, adaptive model coordination, and uncertainty-aware decision-making in increasingly complex agent systems.

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

# A. Proofs

## A.1. Proof of Theorem 3.1

Let $\hat{\lambda}^{(a)}$ denote the calibrated threshold obtained from the calibration set by Eq. (7). For the test sample $(x_{n+1}, y_{n+1}^*)$, let

$$err_{n+1}^{(a)} = \mathbf{1}\left\{A\left(y_{n+1}^*, \hat{y}_{n+1}^{(a)}\right) = 0\right\},$$

$$S_{n+1}^{(a)}(\hat{\lambda}^{(a)}) = \mathbf{1}\left\{u_{n+1}^{(a)} \leq \hat{\lambda}^{(a)}\right\},$$

and

$$Z_{n+1}^{(a)}(\hat{\lambda}^{(a)}) = S_{n+1}^{(a)}(\hat{\lambda}^{(a)}) \cdot err_{n+1}^{(a)}.$$

If $\mathbb{E}\left[S_{n+1}^{(a)}(\hat{\lambda}^{(a)})\right] = 0$, then the calibrated rule accepts a test example with probability zero, and the guarantee is vacuous. We therefore consider the case

$$\mathbb{E}\left[S_{n+1}^{(a)}(\hat{\lambda}^{(a)})\right] > 0.$$

The selection-conditioned error rate of the calibrated rule can be written as

$$\begin{aligned}
&\Pr\left(err_{n+1}^{(a)} = 1 \mid u_{n+1}^{(a)} \leq \hat{\lambda}^{(a)}\right) \\
&= \Pr\left(err_{n+1}^{(a)} = 1 \mid S_{n+1}^{(a)}(\hat{\lambda}^{(a)}) = 1\right) \\
&= \frac{\Pr\left(err_{n+1}^{(a)} = 1, S_{n+1}^{(a)}(\hat{\lambda}^{(a)}) = 1\right)}{\Pr\left(S_{n+1}^{(a)}(\hat{\lambda}^{(a)}) = 1\right)} \\
&= \frac{\mathbb{E}\left[Z_{n+1}^{(a)}(\hat{\lambda}^{(a)})\right]}{\mathbb{E}\left[S_{n+1}^{(a)}(\hat{\lambda}^{(a)})\right]}.
\end{aligned} \tag{12}$$

Thus, it suffices to prove the linear expectation constraint

$$\mathbb{E}\left[Z_{n+1}^{(a)}(\hat{\lambda}^{(a)}) - \alpha S_{n+1}^{(a)}(\hat{\lambda}^{(a)})\right] \leq 0. \tag{13}$$

We now connect the calibration condition to this population-level linear constraint. For any candidate threshold $\lambda^{(a)}$, recall that

$$S_i^{(a)}(\lambda^{(a)}) = \mathbf{1}\left\{u_i^{(a)} \leq \lambda^{(a)}\right\}, \quad Z_i^{(a)}(\lambda^{(a)}) = S_i^{(a)}(\lambda^{(a)})\, err_i^{(a)}.$$

Therefore,

$$Z_i^{(a)}(\lambda^{(a)}) - \alpha S_i^{(a)}(\lambda^{(a)}) = \begin{cases} err_i^{(a)} - \alpha, & \text{if } u_i^{(a)} \leq \lambda^{(a)}, \\ 0, & \text{if } u_i^{(a)} > \lambda^{(a)}. \end{cases}$$

Let $u_{(1)}^{(a)} \leq \cdots \leq u_{(n)}^{(a)}$ be the sorted calibration uncertainty scores and let $err_{(j)}^{(a)}$ denote the corresponding error indicator. For any $\lambda^{(a)}$, define

$$k^{(a)}(\lambda^{(a)}) = \#\{i : u_i^{(a)} \leq \lambda^{(a)}\}.$$

Then the calibration sum can be rewritten as

$$\sum_{i=1}^{n}\left(Z_i^{(a)}(\lambda^{(a)}) - \alpha S_i^{(a)}(\lambda^{(a)})\right) = \sum_{j=1}^{k^{(a)}(\lambda^{(a)})}\left(err_{(j)}^{(a)} - \alpha\right). \tag{14}$$

By the definition of $\hat{\lambda}^{(a)}$ in Eq. (7), the selected threshold satisfies

$$\sum_{j=1}^{k^{(a)}(\hat{\lambda}^{(a)})}\left(err_{(j)}^{(a)} - \alpha\right) \leq -1. \tag{15}$$

Equivalently,

$$\sum_{i=1}^{n} \left( Z_i^{(a)}(\hat{\lambda}^{(a)}) - \alpha S_i^{(a)}(\hat{\lambda}^{(a)}) \right) \leq -1. \tag{16}$$

Following the exchangeability-based leave-one-out calibration argument used in conformal risk control (Angelopoulos et al., 2024), the calibration examples and the test example can be treated symmetrically after applying the finite-sample correction. Since the calibration and test examples are exchangeable, Eq. (16), together with the "+1" correction, implies

$$\mathbb{E}\left[ Z_{n+1}^{(a)}(\hat{\lambda}^{(a)}) - \alpha S_{n+1}^{(a)}(\hat{\lambda}^{(a)}) \right]$$
$$= \frac{1}{n+1} \mathbb{E}\left[ \sum_{i=1}^{n} \left( Z_i^{(a)}(\hat{\lambda}^{(a)}) - \alpha S_i^{(a)}(\hat{\lambda}^{(a)}) \right) + \left( Z_{n+1}^{(a)}(\hat{\lambda}^{(a)}) - \alpha S_{n+1}^{(a)}(\hat{\lambda}^{(a)}) \right) \right]. \tag{17}$$

Substituting Eq. (16) into Eq. (17) gives

$$\mathbb{E}\left[ Z_{n+1}^{(a)}(\hat{\lambda}^{(a)}) - \alpha S_{n+1}^{(a)}(\hat{\lambda}^{(a)}) \right]$$
$$\leq \frac{1}{n+1} \mathbb{E}\left[ -1 + Z_{n+1}^{(a)}(\hat{\lambda}^{(a)}) - \alpha S_{n+1}^{(a)}(\hat{\lambda}^{(a)}) \right]. \tag{18}$$

Since

$$Z_{n+1}^{(a)}(\hat{\lambda}^{(a)}) \leq S_{n+1}^{(a)}(\hat{\lambda}^{(a)})$$

and $S_{n+1}^{(a)}(\hat{\lambda}^{(a)}) \in \{0, 1\}$, we have

$$Z_{n+1}^{(a)}(\hat{\lambda}^{(a)}) - \alpha S_{n+1}^{(a)}(\hat{\lambda}^{(a)}) \leq 1.$$

Therefore,

$$\mathbb{E}\left[ Z_{n+1}^{(a)}(\hat{\lambda}^{(a)}) - \alpha S_{n+1}^{(a)}(\hat{\lambda}^{(a)}) \right] \leq 0. \tag{19}$$

This proves Eq. (13). Rearranging gives

$$\mathbb{E}\left[ Z_{n+1}^{(a)}(\hat{\lambda}^{(a)}) \right] \leq \alpha \, \mathbb{E}\left[ S_{n+1}^{(a)}(\hat{\lambda}^{(a)}) \right].$$

Combining this inequality with Eq. (12), we obtain

$$\Pr\left( err_{n+1}^{(a)} = 1 \mid u_{n+1}^{(a)} \leq \hat{\lambda}^{(a)} \right) \leq \alpha.$$

This establishes the claimed single-model selection-conditioned error control. □

## A.2. Proof of Theorem 3.2

Let $\hat{\boldsymbol{\lambda}} = (\hat{\lambda}^{(a)}, \hat{\lambda}^{(b)})$ denote the calibrated threshold pair obtained from the calibration set by solving Eq. (11). For the test sample $(x_{n+1}, y_{n+1}^*)$, recall the system-level selection indicator

$$S_{n+1}(\hat{\boldsymbol{\lambda}}) \in \{0, 1\},$$

and the system-level accepted-error indicator

$$err_{n+1} \in \{0, 1\}.$$

We define the corresponding joint indicator as

$$Z_{n+1}(\hat{\boldsymbol{\lambda}}) = S_{n+1}(\hat{\boldsymbol{\lambda}}) \cdot err_{n+1}.$$

If $\mathbb{E}\left[ S_{n+1}(\hat{\boldsymbol{\lambda}}) \right] = 0$, then the calibrated routing rule accepts a test example with probability zero, and the guarantee is vacuous. We therefore consider the case

$$\mathbb{E}\left[ S_{n+1}(\hat{\boldsymbol{\lambda}}) \right] > 0.$$

The system-level selection-conditioned error rate can be written as

$$
\Pr\Big(err_{n+1} = 1 \mid S_{n+1}(\hat{\boldsymbol{\lambda}}) = 1\Big)
$$

$$
= \frac{\Pr\Big(err_{n+1} = 1,\, S_{n+1}(\hat{\boldsymbol{\lambda}}) = 1\Big)}{\Pr\Big(S_{n+1}(\hat{\boldsymbol{\lambda}}) = 1\Big)} \tag{20}
$$

$$
= \frac{\mathbb{E}\Big[Z_{n+1}(\hat{\boldsymbol{\lambda}})\Big]}{\mathbb{E}\Big[S_{n+1}(\hat{\boldsymbol{\lambda}})\Big]}.
$$

Thus, it suffices to show

$$
\mathbb{E}\Big[Z_{n+1}(\hat{\boldsymbol{\lambda}}) - \alpha S_{n+1}(\hat{\boldsymbol{\lambda}})\Big] \le 0. \tag{21}
$$

By exchangeability among the calibration and test examples at the level of joint model outputs, $(u_i^{(a)}, u_i^{(b)}, err_i^{(a)}, err_i^{(b)})$, and because the routing policy is deterministic and selects at most one model per input, the induced system-level pairs $\Big(S_i(\hat{\boldsymbol{\lambda}}), Z_i(\hat{\boldsymbol{\lambda}})\Big)$ are exchangeable across examples. Following the same exchangeability-based leave-one-out calibration argument used in conformal risk control (Angelopoulos et al., 2024), we have

$$
\mathbb{E}\Big[Z_{n+1}(\hat{\boldsymbol{\lambda}}) - \alpha S_{n+1}(\hat{\boldsymbol{\lambda}})\Big]
$$

$$
= \frac{1}{n+1}\mathbb{E}\left[\sum_{i=1}^{n}\Big(Z_i(\hat{\boldsymbol{\lambda}}) - \alpha S_i(\hat{\boldsymbol{\lambda}})\Big) + \Big(Z_{n+1}(\hat{\boldsymbol{\lambda}}) - \alpha S_{n+1}(\hat{\boldsymbol{\lambda}})\Big)\right]. \tag{22}
$$

By the definition of the feasible region $\Lambda_\alpha^{(a,b)}$ in Eq. (10), the calibrated pair $\hat{\boldsymbol{\lambda}}$ satisfies

$$
\sum_{i=1}^{n}\Big(Z_i(\hat{\boldsymbol{\lambda}}) - \alpha S_i(\hat{\boldsymbol{\lambda}})\Big) \le -1. \tag{23}
$$

Substituting Eq. (23) into Eq. (22) gives

$$
\mathbb{E}\Big[Z_{n+1}(\hat{\boldsymbol{\lambda}}) - \alpha S_{n+1}(\hat{\boldsymbol{\lambda}})\Big]
$$

$$
\le \frac{1}{n+1}\mathbb{E}\Big[-1 + Z_{n+1}(\hat{\boldsymbol{\lambda}}) - \alpha S_{n+1}(\hat{\boldsymbol{\lambda}})\Big]. \tag{24}
$$

Since routing selects at most one prediction,

$$
Z_{n+1}(\hat{\boldsymbol{\lambda}}) \le S_{n+1}(\hat{\boldsymbol{\lambda}}),
$$

and hence

$$
Z_{n+1}(\hat{\boldsymbol{\lambda}}) - \alpha S_{n+1}(\hat{\boldsymbol{\lambda}}) \le 1.
$$

Therefore,

$$
\mathbb{E}\Big[Z_{n+1}(\hat{\boldsymbol{\lambda}}) - \alpha S_{n+1}(\hat{\boldsymbol{\lambda}})\Big] \le 0. \tag{25}
$$

This proves Eq. (21). Rearranging gives

$$
\mathbb{E}\Big[Z_{n+1}(\hat{\boldsymbol{\lambda}})\Big] \le \alpha\,\mathbb{E}\Big[S_{n+1}(\hat{\boldsymbol{\lambda}})\Big].
$$

Combining this inequality with Eq. (20), we obtain

$$
\Pr\Big(err_{n+1} = 1 \mid S_{n+1}(\hat{\boldsymbol{\lambda}}) = 1\Big) \le \alpha.
$$

Therefore, the two-model routing system satisfies system-level selection-conditioned error control at level $\alpha$. $\qquad\square$

## B. Extension to General Multi-Model Routing Systems

Suppose we have a collection of $M$ foundation models $\{\mathcal{G}^{(1)}, \ldots, \mathcal{G}^{(M)}\}$, where each model $\mathcal{G}^{(m)}$ is equipped with an uncertainty score $u^{(m)}$ and a threshold $\lambda^{(m)}$. Let $\boldsymbol{\lambda} = (\lambda^{(1)}, \ldots, \lambda^{(M)})$ denote the threshold vector. A deterministic routing policy maps the uncertainty scores and thresholds to either a unique accepted model index $r_{\boldsymbol{\lambda}}(x) \in \{1, \ldots, M\}$ or abstention. For example, in a cascaded system, the policy may select the first model whose uncertainty score does not exceed its threshold; if no model satisfies its threshold, the system abstains.

For any fixed threshold vector $\boldsymbol{\lambda}$, the induced system-level selection indicator is

$$S_i(\boldsymbol{\lambda}) = \mathbf{1}\{\text{sample } i \text{ is accepted by one model under } \boldsymbol{\lambda}\}.$$

If $S_i(\boldsymbol{\lambda}) = 1$, the accepted prediction is $\hat{y}_i = \mathcal{G}^{(r_{\boldsymbol{\lambda}}(x_i))}(x_i)$. The corresponding accepted-error indicator is

$$err_i(\boldsymbol{\lambda}) = \mathbf{1}\{S_i(\boldsymbol{\lambda}) = 1 \ \wedge \ A(y_i^*, \hat{y}_i) = 0\}.$$

We then define

$$Z_i(\boldsymbol{\lambda}) = S_i(\boldsymbol{\lambda}) \cdot err_i(\boldsymbol{\lambda}) = err_i(\boldsymbol{\lambda}),$$

where the last equality holds because $err_i(\boldsymbol{\lambda})$ is already defined as an accepted-error indicator.

The system-level selection-conditioned error rate is

$$\text{SCER}(\boldsymbol{\lambda}) = \frac{\mathbb{E}[Z(\boldsymbol{\lambda})]}{\mathbb{E}[S(\boldsymbol{\lambda})]},$$

whenever $\mathbb{E}[S(\boldsymbol{\lambda})] > 0$. Thus, controlling $\text{SCER}(\boldsymbol{\lambda}) \leq \alpha$ is equivalent to the linear expectation constraint

$$\mathbb{E}[Z(\boldsymbol{\lambda}) - \alpha S(\boldsymbol{\lambda})] \leq 0.$$

Following the same finite-sample calibration argument as in the single-model and two-model cases, a sufficient empirical condition is

$$\sum_{i=1}^{n} (Z_i(\boldsymbol{\lambda}) - \alpha S_i(\boldsymbol{\lambda})) \leq -1. \tag{26}$$

Equivalently, since $Z_i(\boldsymbol{\lambda}) = S_i(\boldsymbol{\lambda}) \cdot err_i(\boldsymbol{\lambda})$, Eq. (26) can be written as

$$\sum_{i=1}^{n} (S_i(\boldsymbol{\lambda}) \, err_i(\boldsymbol{\lambda}) - \alpha S_i(\boldsymbol{\lambda})) \leq -1.$$

We define the feasible threshold region as

$$\Lambda_{\alpha}^{(1:M)} = \left\{ \boldsymbol{\lambda} : \sum_{i=1}^{n} (Z_i(\boldsymbol{\lambda}) - \alpha S_i(\boldsymbol{\lambda})) \leq -1 \right\}. \tag{27}$$

Among all feasible threshold vectors, we select the retention-maximizing solution

$$\hat{\boldsymbol{\lambda}} = \underset{\boldsymbol{\lambda} \in \Lambda_{\alpha}^{(1:M)}}{\arg\max} \frac{1}{n} \sum_{i=1}^{n} S_i(\boldsymbol{\lambda}). \tag{28}$$

By the same exchangeability-based calibration argument as above, the resulting routing system satisfies

$$\Pr\left(err_{n+1}(\hat{\boldsymbol{\lambda}}) = 1 \mid S_{n+1}(\hat{\boldsymbol{\lambda}}) = 1\right) \leq \alpha,$$

provided that the routing policy deterministically maps each input to at most one accepted model output or to abstention.

Therefore, LEC extends naturally to routing systems with an arbitrary number of models. The main algorithmic challenge is computational: the threshold vector $\boldsymbol{\lambda}$ is multi-dimensional, and finding the retention-maximizing feasible vector may require an efficient structured search over the threshold space. Nevertheless, this extension does not alter the underlying statistical form of the guarantee, because the system still induces binary selection and accepted-error indicators. More broadly, the same linear expectation transformation can apply to other task-specific ratio-form risk metrics, as long as the numerator and denominator can be represented through suitable system-level indicators.

# C. Additional Experimental Settings

**Details of Utilized Datasets and Models.** For the closed-ended CommonsenseQA dataset, we employ both the full training split (9,741 samples) and the validation split (1,221 samples)[1]. We remove a small number of samples containing non-ASCII characters in either the query or answer that cannot be encoded by the tokenizer. From the remaining data, we select one QA pair as a fixed one-shot demonstration, which is prepended to the prompt for all other samples. After filtering and prompt construction, we select 10,000 QA instances in total. An example of the complete prompt is presented as follows:

```
### System:
Make your best effort and select the correct answer for the following
multiple-choice question.  For each question, only one choice is correct.  Answer
should be one among A, B, C, D, E.

### User:
What is something I need to avoid while playing ball?
A: competition
B: losing
C: injury
D: hitting the ball
E: having fun
### Assistant:
C

### User:
The sanctions against the school were a punishing blow, and they seemed to what the
efforts the school had made to change?
A: ignore
B: enforce
C: authoritarian
D: yell at
E: avoid
### Assistant:
```

For the open-ended TriviaQA dataset, we randomly select 8,000 QA pairs from the validation split of the `rc.nocontext` subset[2]. We also apply a one-shot prompt for each data point. An example of the complete prompt is presented as follows:

```
### System:
This is a bot that correctly answers questions.

### User:
In 1968, who did radical feminist Valerie Solanas shoot and wound as he entered his
New York studio?
### Assistant:
Andy Warhol

### User:
Who was the man behind The Chipmunks?
### Assistant:
```

For the open-ended MM-Vet v2 dataset, we adopt the total test split (517 VQA samples) [3] for evaluation. An example of the complete prompt is presented as follows:

```
<image>
What is x in the equation?
NOTE: Provide only the final answer.  Do not provide unrelated details.
```

---

[1]Source files of the CommonsenseQA dataset.
[2]Source files of the TriviaQA dataset.
[3]Source files of the MM-Vet v2 dataset.

For the closed-ended ScienceQA dataset, we utilize the test split (4,241 samples)[4]. Due to missing visual inputs in a subset of the data, we retain 2,017 VQA samples for evaluation. An example of the complete prompt is presented as follows:

```
<image>
Which of the following could Gordon's test show?
A: if the spacecraft was damaged when using a parachute with a 1 m vent going 200 km
per hour
B: how steady a parachute with a 1 m vent was at 200 km per hour
C: whether a parachute with a 1 m vent would swing too much at 400 km per hour

This is a single choice question, answer only with choice number in A, B, C.
```

In QA tasks, we employ four series of open-source LLMs available on Hugging Face: OpenChat, LLaMA, Vicuna, and Qwen, divided by the model size into: (1) 3B: Qwen-2.5-3B-Instruct; (2) 7B: OpenChat-3.5, Vicuna-7B-v1.5, and Qwen-2.5-7B-Instruct; (3) 8B: LLaMA-3.1-8B-Instruct; (4) 13B: Vicuna-13B-v1.5; (5) 14B: Qwen-2.5-14B-Instruct; (6) 70B: LLaMA-3.1-70B-Instruct. In VQA tasks, we employ three distinct LVLM groups: LLaVA1.5, LLaVA-NeXT, and InternVL2, divided by the model size into: (1) 1B: InternVL2-1B; (2) 7B: LLaVA-1.5-7B-HF and LLaVA-V1.6-Mistral-7B-HF; (3) 8B: InternVL2-8B. We omit "-Instruct" and "-HF" when reporting the experimental results.

Due to imperfect instruction-following behavior in some models, a very small number of predictions may be invalid and thus removed during preprocessing. Specifically, in closed-ended tasks, a few model outputs do not strictly follow the prescribed option format, while in open-ended tasks, rare cases may result in empty responses after standard cleaning and post-processing. Consequently, the set of valid evaluation samples can differ slightly across models, although the number of excluded samples is negligible. In the two-model routing setting, we therefore restrict evaluation to the subset of samples that are shared by both models to ensure a fair and consistent comparison. As shown in Table 2, the set of evaluation samples shared between LLaMA-3.1-8B and the two Qwen primary models exhibits a minor discrepancy, stemming from a very small number of invalid predictions that are removed during preprocessing. This difference is negligible in scale and does not materially affect the evaluation or the conclusions regarding the effectiveness of the proposed method.

**Details of Alignment Criteria.** In closed-ended QA or VQA tasks, we can directly determine whether the predicted option is consistent with the ground-truth option. In open-ended settings, following previous work (Duan et al., 2024; Wang et al., 2026), we estimate the sentence similarity between two answers (ground truth, the most likely answer, or sampled answer) leveraging SentenceTransformers (Reimers & Gurevych, 2019a) with DistillRoBERTa (Sanh et al., 2019) as the backbone. For bi-entailment (Kuhn et al., 2023; Farquhar et al., 2024; Wang et al., 2025c), we employ DeBERTa-v3[5] as the Natural Language Inference (NLI) classifier, which outputs logits over three semantic relation classes: entailment, neutral, and contradiction. Two answers are deemed semantically aligned if the classifier predicts entailment for both directions. In addition, we also adopt LLM-as-a-Judge by prompting the Qwen2.5-7B model with the following instruction:

```
You are an expert evaluator for open-ended QA correctness.

Given a question, a ground-truth answer, and a model's answer, decide which option
best describes the model's answer:
A. correct { semantically equivalent to the ground-truth answer.
B. partial { related and contains some correct information but is incomplete or
partially wrong.
C. incorrect { not compatible with the ground-truth answer.

Respond by selecting exactly one of A, B, or C.

Question:  <TEXT>
Ground truth answer:  <TEXT>
Model answer:  <TEXT>

Answer:
```

---

[4]Source files of the ScienceQA dataset.

[5]We use DeBERTa-v3-large-mnli-fever-anli-ling-wanli. Different from DeBERTa-large (He et al., 2021) employed in previous research (Lin et al., 2024), the three dimensions of its output logits correspond to entailment, neutral, and contradiction, respectively.

We adopt the partial correctness criterion by default. Note that the statistical validity of our `LEC` framework is not affected by changes in the alignment criterion of admission function $A$.

**Details of Uncertainty Estimators.** In the closed-ended CommonsenseQA dataset, we compute the PE as $\sum_o -p_o \log p_o$, where $p_o$ is the probability of the $o$-th option. In black-box settings, we sample additional 20 answers (i.e., options) and utilize the normalized frequency score as $p_o$. We only compute black-box PE in the closed-ended ScienceQA (VQA) dataset. In the open-ended TriviaQA (QA) and MM-Vet v2 (VQA) datasets, we sample additional 10 answers by default to compute black-box SE, EigV, Ecc, and Deg. In black-box SE, we perform semantic clustering via the bi-entailment criterion. See Farquhar et al. (2024) for more details of black-box SE. See Lin et al. (2024) for more details of EigV, Ecc, and Deg. For SELF, we compute the length-normalized sentence entropy of the most likely generation. See Duan et al. (2024) for details.

**Details of Additional Hyperparameters.** In black-box settings, we set the sampling temperature to 1.0 and top-p to 0.9. For both the CommonsenseQA and ScienceQA datasets, limit the model's output to a single token, since only the option letter is required. For the TriviaQA dataset, we set the maximum output length to 36 tokens. For the MM-Vet v2 dataset, we set the maximum output length to 32 tokens. In UCB-based calibration, we set the significance level $\delta$ to 0.05.

**Details of Baselines.** As presented in Algorithms 1 and 2, we detail the threshold calibration procedures in single-model selective prediction for two UCB-based approaches and the proposed `LEC`. Algorithm 3 details `LEC-Routing` that follows naturally by reformulating the selection and joint indicators at the system level, without altering the overall calibration procedure. Moreover, in the two-model routing setting, rather than selecting the largest feasible threshold pair, we adopt the threshold pair that yields the maximum number of accepted samples. See Jung et al. (2025) for UCB-based routing.

---

**Algorithm 1** UCB-based threshold calibration for single-model selective prediction with PAC-style accepted-error control

1: **Input:** Primary model $\mathcal{G}^{(a)}$, calibration set $\{(x_i, y_i^*, \hat{y}_i^{(a)})\}_{i=1}^N$, uncertainty estimator $\mathcal{U}$, admission function $A$, risk level $\alpha$, significance level $\delta$, confidence interval type (HFD or CLP)
2: **Output:** Calibrated threshold $\hat{\lambda}^{(a)}$
3: Compute uncertainty scores for all $i \in [N]$: $u_i^{(a)} \leftarrow \mathcal{U}(x_i; \mathcal{G}^{(a)})$;
4: Compute error indicators for all $i \in [N]$: $err_i^{(a)} \leftarrow \mathbf{1}\{A(y_i^*, \hat{y}_i^{(a)}) = 0\}$;
5: Sort uncertainty scores in ascending order $u_{(1)}^{(a)} \leq \cdots \leq u_{(N)}^{(a)}$, with corresponding error indicators $err_{(1)}^{(a)}, \ldots, err_{(N)}^{(a)}$;
6: Initialize $\hat{\lambda}^{(a)} \leftarrow$ NULL;
7: **for** $i = 1$ **to** $N$ **do**
8:      Let $\lambda \leftarrow u_{(i)}^{(a)}$;
9:      Number of accepted samples: $n_\lambda \leftarrow \sum_{j=1}^N \mathbf{1}\{u_{(j)}^{(a)} \leq \lambda\}$;
10:      Number of errors among accepted samples: $X_\lambda \leftarrow \sum_{j=1}^N \mathbf{1}\{u_{(j)}^{(a)} \leq \lambda \wedge err_{(j)}^{(a)} = 1\}$;
11:      **if** confidence interval type is HFD **then**
12:          Compute Hoeffding-style $(1 - \delta)$ upper confidence bound: $\text{UCB} \leftarrow \frac{X_\lambda}{n_\lambda} + \sqrt{\frac{\log(1/\delta)}{2n_\lambda}}$;
13:      **else if** confidence interval type is CLP **then**
14:          **if** $X_\lambda = n_\lambda$ **then**
15:              Set $\text{UCB} \leftarrow 1$;
16:          **else**
17:              Compute Clopper-Pearson-style $(1-\delta)$ upper confidence bound: $\text{UCB} \leftarrow \text{BetaInv}(1 - \delta; X_\lambda + 1, n_\lambda - X_\lambda)$;
18:          **end if**
19:      **end if**
20:      **if** $\text{UCB} \leq \alpha$ **then**
21:          Update $\hat{\lambda}^{(a)} \leftarrow \lambda$;
22:      **end if**
23: **end for**
24: **if** $\hat{\lambda}^{(a)} =$ NULL **then**
25:      **Return** "No feasible threshold for risk level $\alpha$"
26: **else**
27:      **Return** $\hat{\lambda}^{(a)}$
28: **end if**

---

---

**Algorithm 2** Threshold calibration via `LEC` for single-model selective prediction with selection-conditioned error control

1: **Input:** $\mathcal{G}^{(a)}$, $\{(x_i, y_i^*, \hat{y}_i^{(a)})\}_{i=1}^N$, $\mathcal{U}$, $A$, $\alpha$
2: **Output:** Calibrated threshold $\hat{\lambda}^{(a)}$
3: Compute uncertainty scores and error indicators for all $i \in [N]$: $u_i^{(a)} \leftarrow \mathcal{U}(x_i; \mathcal{G}^{(a)})$, $err_i^{(a)} \leftarrow \mathbf{1}\{A(y_i^*, \hat{y}_i^{(a)}) = 0\}$;
4: Sort uncertainty scores in ascending order $u_{(1)}^{(a)} \leq \cdots \leq u_{(N)}^{(a)}$, with corresponding error indicators $err_{(1)}^{(a)}, \ldots, err_{(N)}^{(a)}$;
5: Define candidate set $\mathcal{T}^{(a)} \leftarrow \{u_{(1)}^{(a)}, \ldots, u_{(N)}^{(a)}\}$;
6: Initialize $\hat{\lambda}^{(a)} \leftarrow$ NULL;
7: **for** each $\lambda \in \mathcal{T}^{(a)}$ **do**
8:      Obtain selection indicators for all $i \in [N]$: $S_i^{(a)}(\lambda) \leftarrow \mathbf{1}\{u_i^{(a)} \leq \lambda\}$;
9:      Obtain joint indicators for all $i \in [N]$: $Z_i^{(a)}(\lambda) \leftarrow S_i^{(a)}(\lambda) \cdot err_i^{(a)}$;
10:      Compute the empirical linear constraint: $L(\lambda) \leftarrow \sum_{i=1}^N \left( Z_i^{(a)}(\lambda) - \alpha S_i^{(a)}(\lambda) \right)$;
11:      **if** $L(\lambda) \leq -1$ **then**
12:          Update $\hat{\lambda}^{(a)} \leftarrow \lambda$;
13:      **end if**
14: **end for**
15: **if** $\hat{\lambda}^{(a)} =$ NULL **then**
16:      **Return** "No feasible threshold for risk level $\alpha$";
17: **else**
18:      **Return** $\hat{\lambda}^{(a)}$;
19: **end if**

---

**Algorithm 3** Joint threshold calibration via `LEC-Routing` for two-model routing systems

1: **Input:** Models $(\mathcal{G}^{(a)}, \mathcal{G}^{(b)})$, calibration set $\{(u_i^{(a)}, u_i^{(b)}, err_i^{(a)}, err_i^{(b)})\}_{i=1}^N$, risk level $\alpha$
2: **Output:** Calibrated threshold pair $(\hat{\lambda}^{(a)}, \hat{\lambda}^{(b)})$
3: Sort $\{u_i^{(a)}\}_{i=1}^N$ in ascending order and define candidate set $\mathcal{T}^{(a)} \leftarrow \{u_{(1)}^{(a)}, \ldots, u_{(N)}^{(a)}\}$;
4: Sort $\{u_i^{(b)}\}_{i=1}^N$ in ascending order and define candidate set $\mathcal{T}^{(b)} \leftarrow \{u_{(1)}^{(b)}, \ldots, u_{(N)}^{(b)}\}$;
5: Initialize feasible set $\Lambda_\alpha^{(a,b)} \leftarrow \emptyset$;
6: **for** each $\lambda^{(a)} \in \mathcal{T}^{(a)}$ **do**
7:      **for** each $\lambda^{(b)} \in \mathcal{T}^{(b)}$ **do**
8:          Obtain system-level selection indicators: $S_i(\lambda^{(a)}, \lambda^{(b)}) \leftarrow \mathbf{1}\{u_i^{(a)} \leq \lambda^{(a)}\} + \mathbf{1}\{u_i^{(a)} > \lambda^{(a)} \wedge u_i^{(b)} \leq \lambda^{(b)}\}$;
9:          Obtain system-level joint indicators: $Z_i(\lambda^{(a)}, \lambda^{(b)}) \leftarrow \mathbf{1}\{u_i^{(a)} \leq \lambda^{(a)} \wedge err_i^{(a)} = 1\} + \mathbf{1}\{u_i^{(a)} > \lambda^{(a)} \wedge u_i^{(b)} \leq \lambda^{(b)} \wedge err_i^{(b)} = 1\}$;
10:          Compute the system-level empirical linear constraint: $L(\lambda^{(a)}, \lambda^{(b)}) \leftarrow \sum_{i=1}^N \left( Z_i(\lambda^{(a)}, \lambda^{(b)}) - \alpha S_i(\lambda^{(a)}, \lambda^{(b)}) \right)$;
11:          **if** $L(\lambda^{(a)}, \lambda^{(b)}) \leq -1$ **then**
12:              Add $(\lambda^{(a)}, \lambda^{(b)})$ to $\Lambda_\alpha^{(a,b)}$;
13:          **end if**
14:      **end for**
15: **end for**
16: **if** $\Lambda_\alpha^{(a,b)} = \emptyset$ **then**
17:      **Return** "No feasible threshold pair for risk level $\alpha$";
18: **else**
19:      Select the retention-maximizing feasible threshold pair: $(\hat{\lambda}^{(a)}, \hat{\lambda}^{(b)}) \leftarrow \operatorname{argmax}_{(\lambda^{(a)}, \lambda^{(b)}) \in \Lambda_\alpha^{(a,b)}} \sum_{i=1}^N S_i(\lambda^{(a)}, \lambda^{(b)})$;
20:      **Return** $(\hat{\lambda}^{(a)}, \hat{\lambda}^{(b)})$.
21: **end if**

---

When iterating over candidate thresholds, one can adopt binary search or other accelerated strategies. The search space can be adjusted according to the specific UQ method, and finer threshold granularity can be achieved by decreasing the search step size. All such procedures are performed offline during calibration; the test-time deployment remains fully real-time.

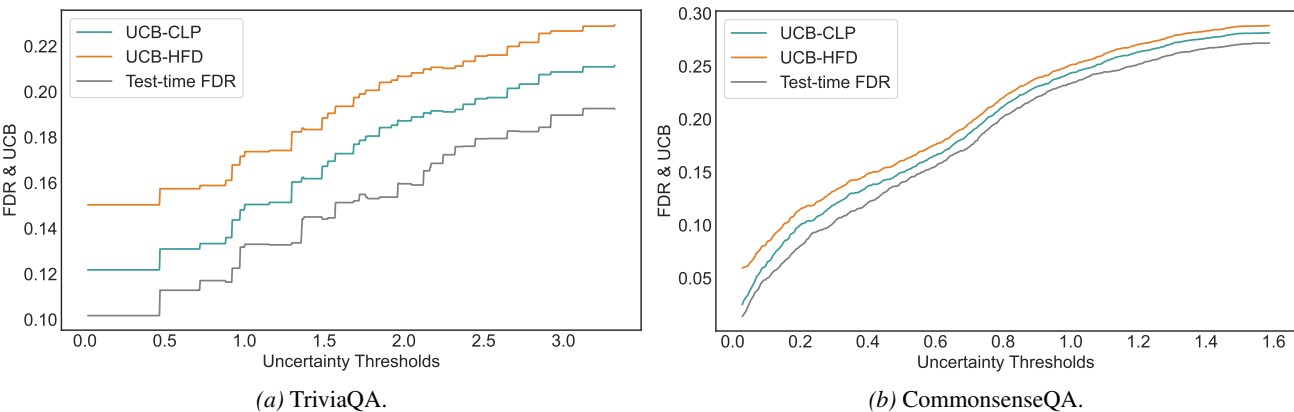

*Figure 9.* Two styles of UCBs versus the test-time empirical selection-conditioned error rate at various uncertainty thresholds. In (a), we use the LLaMA-3.1-8B model with white-box PE as the uncertainty estimator; in (b), we use Qwen2.5-14B with SE as the uncertainty estimator. The y-axis label "FDR" denotes the observed fraction of erroneous predictions among accepted predictions.

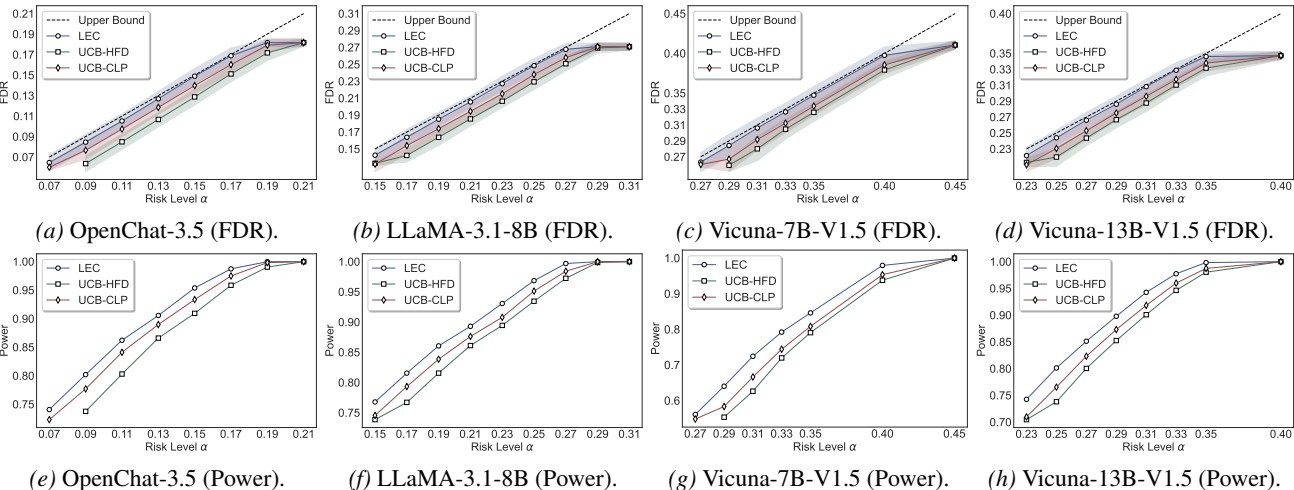

*Figure 10.* Test-time empirical selection-conditioned error rate (mean±std) and power (mean) on CommonsenseQA using black-box PE.

# D. Additional Experimental Results

**Inherent Conservation of Confidence Intervals.** Figure 9 illustrates the gap between the test-time selection-conditioned error rate and the UCBs computed on the calibration set across a range of uncertainty thresholds. Specifically, Figure 9 (a) reports results on TriviaQA using LLaMA-3.1-8B with white-box PE, while Figure 9 (b) shows results on CommonsenseQA using Qwen2.5-14B with SE. A key empirical observation is that, across almost all uncertainty thresholds, both Hoeffding-style UCBs and exact Clopper–Pearson UCBs systematically overestimate the corresponding test-time selection-conditioned error rate, indicating that the conservativeness is not merely due to loose concentration inequalities, but rather reflects a more fundamental limitation of UCB-based calibration. This behavior can be attributed to the objective of UCB-based methods, which aim to control worst-case tail events with high probability. To satisfy this requirement, the UCB must remain valid even under rare but adversarial realizations of the calibration data, such as observing an unusually low empirical error rate despite a relatively high underlying risk.

**Evaluation of Statistical Validity and Power on CommonsenseQA at Black-Box Settings.** Figure 10 reports the test-time selection-conditioned error rate and power on the CommonsenseQA dataset using black-box PE across multiple LLMs. As presented in Figures 10 (a)–(d), LEC consistently achieves valid selection-conditioned risk control at test time, with the realized selection-conditioned error rate remaining below the target risk level across all models and risk settings. Despite the increased noise and reduced resolution typically associated with black-box uncertainty estimates, the proposed framework preserves its finite-sample guarantees, indicating that LEC does not rely on privileged model information. In terms of power,

*Table 3.* Power comparison on the CommonsenseQA dataset (mean).

| LLMs | Methods / $\alpha$ | 0.05 | 0.1 | 0.15 | 0.2 | 0.25 | 0.3 | 0.35 | 0.4 | 0.45 |
|---|---|---|---|---|---|---|---|---|---|---|
| OpenChat-3.5 | UCB-CLP | 0.6556 | 0.8397 | 0.9401 | $0.9999_{(1)}$ | $0.9999_{(4)}$ | $0.9999_{(4)}$ | $0.9999_{(4)}$ | $0.9999_{(4)}$ | $0.9999_{(4)}$ |
| | UCB-HFD | 0.4887 | 0.8024 | 0.9214 | 0.9989 | $0.9999_{(4)}$ | $0.9999_{(4)}$ | $0.9999_{(4)}$ | $0.9999_{(4)}$ | $0.9999_{(4)}$ |
| | LEC | **0.6850** | **0.8559** | **0.9567** | $\mathbf{0.9999}_{(4)}$ | $\mathbf{0.9999}_{(4)}$ | $\mathbf{0.9999}_{(4)}$ | $\mathbf{0.9999}_{(4)}$ | $\mathbf{0.9999}_{(4)}$ | $\mathbf{0.9999}_{(4)}$ |
| Qwen2.5-3B | UCB-CLP | 0.1803 | 0.6242 | 0.8084 | 0.9114 | 0.9964 | $0.9999_{(9)}$ | $0.9999_{(9)}$ | $0.9999_{(9)}$ | $0.9999_{(9)}$ |
| | UCB-HFD | - | 0.4976 | 0.7758 | 0.8968 | 0.9851 | $0.9999_{(9)}$ | $0.9999_{(9)}$ | $0.9999_{(9)}$ | $0.9999_{(9)}$ |
| | LEC | **0.2553** | **0.6779** | **0.8366** | **0.9315** | $\mathbf{0.9999}_{(8)}$ | $\mathbf{0.9999}_{(9)}$ | $\mathbf{0.9999}_{(9)}$ | $\mathbf{0.9999}_{(9)}$ | $\mathbf{0.9999}_{(9)}$ |
| Qwen2.5-7B | UCB-CLP | 0.5704 | 0.7647 | 0.8810 | 0.9685 | $0.9999_{(8)}$ | $0.9999_{(8)}$ | $0.9999_{(8)}$ | $0.9999_{(8)}$ | $0.9999_{(8)}$ |
| | UCB-HFD | 0.3546 | 0.7186 | 0.8573 | 0.9522 | $0.9999_{(8)}$ | $0.9999_{(8)}$ | $0.9999_{(8)}$ | $0.9999_{(8)}$ | $0.9999_{(8)}$ |
| | LEC | **0.6108** | **0.7850** | **0.8982** | **0.9839** | $\mathbf{0.9999}_{(8)}$ | $\mathbf{0.9999}_{(8)}$ | $\mathbf{0.9999}_{(8)}$ | $\mathbf{0.9999}_{(8)}$ | $\mathbf{0.9999}_{(8)}$ |
| Qwen2.5-14B | UCB-CLP | 0.6583 | 0.8644 | 0.9595 | $0.9999_{(7)}$ | $0.9999_{(7)}$ | $0.9999_{(7)}$ | $0.9999_{(7)}$ | $0.9999_{(7)}$ | $0.9999_{(7)}$ |
| | UCB-HFD | 0.4402 | 0.8320 | 0.9426 | $0.9999_{(7)}$ | $0.9999_{(7)}$ | $0.9999_{(7)}$ | $0.9999_{(7)}$ | $0.9999_{(7)}$ | $0.9999_{(7)}$ |
| | LEC | **0.6997** | **0.8833** | **0.9746** | $\mathbf{0.9999}_{(7)}$ | $\mathbf{0.9999}_{(7)}$ | $\mathbf{0.9999}_{(7)}$ | $\mathbf{0.9999}_{(7)}$ | $\mathbf{0.9999}_{(7)}$ | $\mathbf{0.9999}_{(7)}$ |
| Vicuna-7B-V1.5 | UCB-CLP | - | 0.0346 | 0.1039 | 0.3437 | 0.5053 | 0.6684 | 0.8283 | 0.9641 | $0.9999_{(9)}$ |
| | UCB-HFD | - | - | 0.0583 | 0.2717 | 0.4674 | 0.6312 | 0.7976 | 0.9517 | $0.9999_{(9)}$ |
| | LEC | - | **0.0575** | **0.2263** | **0.3963** | **0.5477** | **0.7099** | **0.8716** | **0.9845** | $\mathbf{0.9999}_{(9)}$ |
| Vicuna-13B-V1.5 | UCB-CLP | - | 0.1365 | 0.4695 | 0.6468 | 0.7945 | 0.9138 | 0.9901 | $0.9999_{(9)}$ | $0.9999_{(9)}$ |
| | UCB-HFD | - | - | 0.3829 | 0.6188 | 0.7643 | 0.8989 | 0.9818 | $0.9999_{(9)}$ | $0.9999_{(9)}$ |
| | LEC | **0.0543** | **0.2785** | **0.5278** | **0.6736** | **0.8311** | **0.9342** | **0.9989** | $\mathbf{0.9999}_{(9)}$ | $\mathbf{0.9999}_{(9)}$ |
| LLaMA-3.1-8B | UCB-CLP | 0.3903 | 0.6078 | 0.7702 | 0.8739 | 0.9556 | $0.9999_{(7)}$ | $0.9999_{(7)}$ | $0.9999_{(7)}$ | $0.9999_{(7)}$ |
| | UCB-HFD | - | 0.5502 | 0.7398 | 0.8596 | 0.9417 | $0.9999_{(6)}$ | $0.9999_{(7)}$ | $0.9999_{(7)}$ | $0.9999_{(7)}$ |
| | LEC | **0.4341** | **0.6468** | **0.7939** | **0.8893** | **0.9745** | $\mathbf{0.9999}_{(7)}$ | $\mathbf{0.9999}_{(7)}$ | $\mathbf{0.9999}_{(7)}$ | $\mathbf{0.9999}_{(7)}$ |
| LLaMA-3.1-70B | UCB-CLP | 0.6328 | 0.8099 | 0.9193 | 0.9929 | $0.9999_{(7)}$ | $0.9999_{(7)}$ | $0.9999_{(7)}$ | $0.9999_{(7)}$ | $0.9999_{(7)}$ |
| | UCB-HFD | 0.5109 | 0.7792 | 0.9012 | 0.9856 | $0.9999_{(7)}$ | $0.9999_{(7)}$ | $0.9999_{(7)}$ | $0.9999_{(7)}$ | $0.9999_{(7)}$ |
| | LEC | **0.6570** | **0.8323** | **0.9383** | **0.9990** | $\mathbf{0.9999}_{(7)}$ | $\mathbf{0.9999}_{(7)}$ | $\mathbf{0.9999}_{(7)}$ | $\mathbf{0.9999}_{(7)}$ | $\mathbf{0.9999}_{(7)}$ |

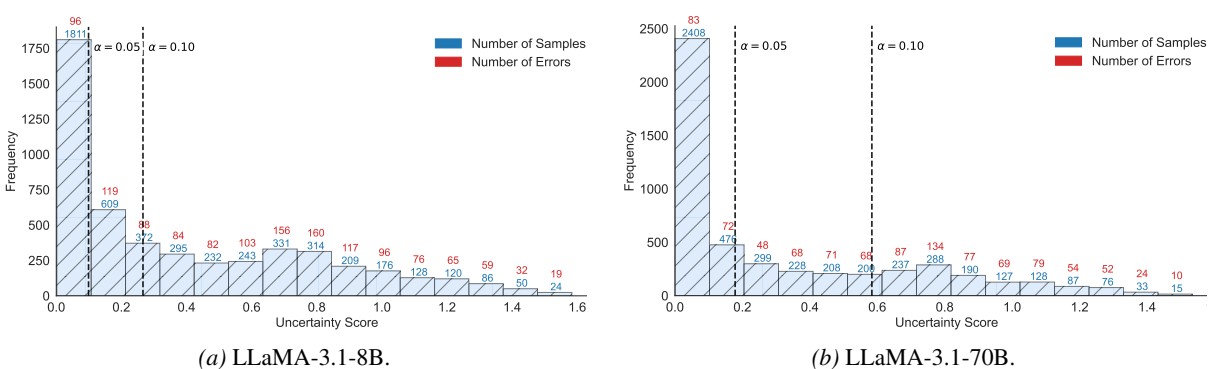

*(a)* LLaMA-3.1-8B.            *(b)* LLaMA-3.1-70B.

*Figure 11.* Uncertainty and correctness distribution on the CommonsenseQA dataset.

Figures 10 (e)–(h) demonstrate that LEC consistently retains more admissible samples than UCB-based baselines under the same risk constraints. These results confirm that the advantages of LEC extend naturally to black-box settings, where only limited uncertainty information is available.

**Power Analysis on CommonsenseQA.** Table 3 reports a comprehensive comparison of power on CommonsenseQA across eight LLMs and various risk levels. Consistent with the main results, LEC uniformly achieves higher power than both UCB-based baselines under the same target risk constraints. This trend holds across all evaluated models and becomes particularly pronounced at low to moderate risk levels, where conservative calibration has the largest impact on sample retention. Notably, for several models, including Vicuna-13B-V1.5, LEC remains feasible at substantially lower risk levels where UCB-based methods either yield significantly lower power or fail to identify valid thresholds. This further highlights the advantage of expectation-level risk control in retaining admissible samples under strict reliability requirements.

**Uncertainty-Correctness Distribution.** Figure 11 visualizes the joint distribution of uncertainty scores and correctness on the test set for LLaMA-3.1-8B and LLaMA-3.1-70B. Each histogram bin reports the number of test samples within a given uncertainty range, along with the corresponding number of incorrect predictions. A key observation is that incorrect

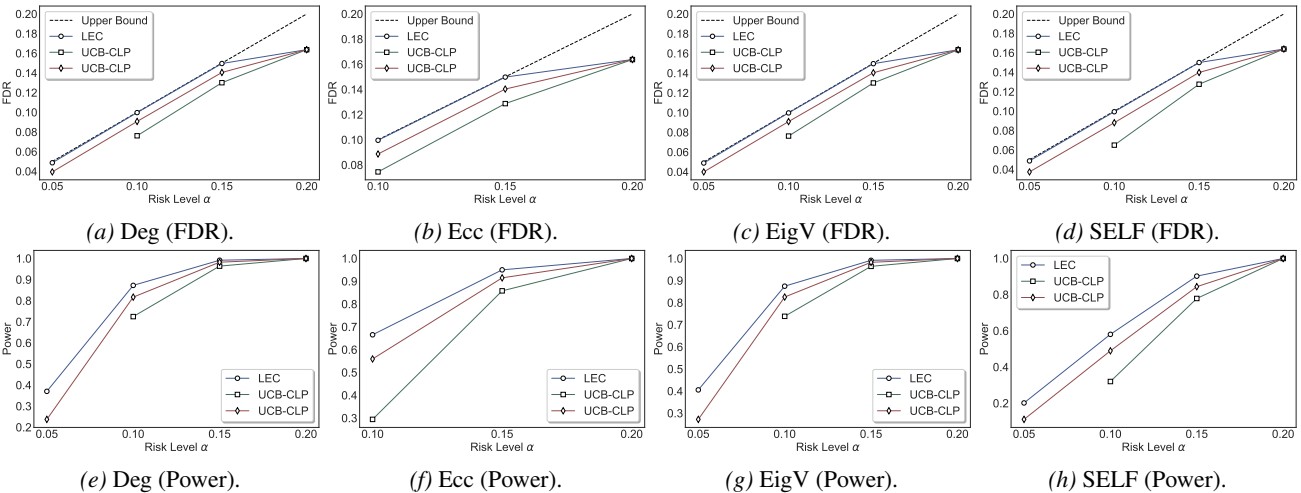

*(a) Deg (FDR).*  *(b) Ecc (FDR).*  *(c) EigV (FDR).*  *(d) SELF (FDR).*

*(e) Deg (Power).*  *(f) Ecc (Power).*  *(g) EigV (Power).*  *(h) SELF (Power).*

*Figure 12.* Test-time empirical selection-conditioned error rate and power on the TriviaQA dataset with the Qwen2.5-3B model using different UQ methods (mean). `LEC` provides tighter risk control while retaining more correct samples.

predictions are present across nearly all uncertainty intervals, rather than being confined to a small high-uncertainty region. This indicates that selective prediction inherently involves a trade-off between rejecting uncertain samples and retaining correct ones, and that no single uncertainty threshold can perfectly separate correct and incorrect predictions.

Importantly, the two models exhibit markedly different uncertainty-correctness profiles. Compared to LLaMA-3.1-8B, LLaMA-3.1-70B assigns lower uncertainty scores to a larger fraction of correct predictions, while maintaining a comparable or lower error density in the low-uncertainty region. As a result, for the same target risk level, the larger model can accept a greater number of correct samples before violating the risk constraint, leading to consistently higher power. These results reinforce that the performance gains of `LEC` are driven not only by tighter calibration, but also by its ability to adapt to model-specific uncertainty–correctness characteristics. By directly constraining expected system-level risk, `LEC` effectively leverages favorable uncertainty distributions to retain more admissible samples, while preserving rigorous statistical validity.

**Robustness and Efficiency across UQ Methods and Sampling Sizes on TriviaQA.** Figure 12 evaluates `LEC` on TriviaQA using four different uncertainty estimators (Deg, Ecc, EigV, and SELF). Across all uncertainty methods, `LEC` consistently achieves tighter risk control and higher power than UCB-based baselines, demonstrating that the benefits of `LEC` are orthogonal to the specific choice of uncertainty estimator. Table 4 further examines the effect of sampling size for SE and EigV uncertainty estimators on LLaMA-3.1-70B. Even with as few as 5 samples, `LEC` maintains valid risk control, highlighting its strong finite-sample efficiency. As the sampling size increases, both AUROC and power improve monotonically, while selection-conditioned error rate remains tightly controlled. This trend confirms that `LEC` is able to effectively translate improvements in uncertainty quality into tangible gains in selective prediction performance, without requiring large calibration budgets.

*Table 4.* Test-time empirical selection-conditioned error rate and power on TriviaQA with the LLaMA-3.1-70B model across various user-specified risk levels under different uncertainty methods and sampling sizes (mean). We also report AUROC to reflect the quality of UQ methods.

| Uncertainty Estimators | Sampling Sizes | Test-time FDR | | | | Power | | | | AUROC |
|---|---|---|---|---|---|---|---|---|---|---|
| | | **0.02** | **0.03** | **0.04** | **0.05** | **0.02** | **0.03** | **0.04** | **0.05** | |
| SE | 5 | 0.0157 | 0.0266 | 0.0362 | 0.0470 | 0.7573 | 0.9326 | 0.9730 | 0.9985 | 0.8007 |
| | 10 | 0.0193 | 0.0296 | 0.0396 | 0.0474 | 0.8667 | 0.9491 | 0.9831 | 0.9995 | 0.8247 |
| | 15 | 0.0197 | 0.0299 | 0.0398 | 0.0474 | 0.8903 | 0.9545 | 0.9851 | 0.9995 | 0.8379 |
| | 20 | 0.0198 | 0.0300 | 0.0399 | 0.0474 | 0.8930 | 0.9551 | 0.9867 | 0.9995 | 0.8400 |
| EigV | 5 | 0.0198 | 0.0299 | 0.0398 | 0.0475 | 0.9364 | 0.9794 | 0.9949 | 0.9996 | 0.8668 |
| | 10 | 0.0199 | 0.0299 | 0.0397 | 0.0475 | 0.9414 | 0.9804 | 0.9960 | 0.9996 | 0.8821 |
| | 15 | 0.0198 | 0.0299 | 0.0398 | 0.0475 | 0.9430 | 0.9817 | 0.9965 | 0.9996 | 0.8853 |
| | 20 | 0.0199 | 0.0299 | 0.0397 | 0.0474 | 0.9436 | 0.9829 | 0.9968 | 0.9997 | 0.8874 |

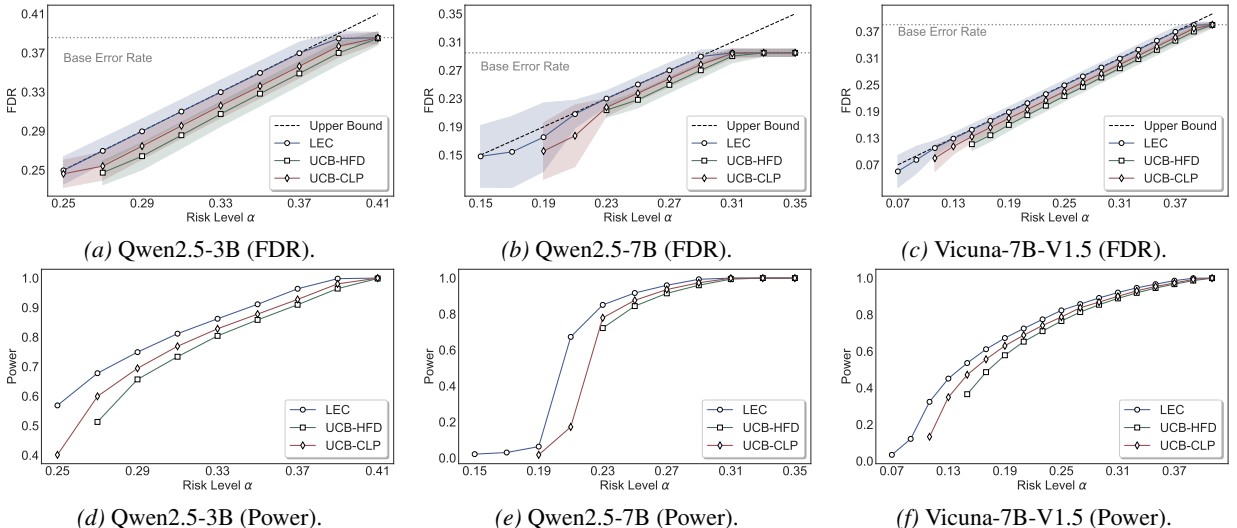

*(a)* Qwen2.5-3B (FDR).  *(b)* Qwen2.5-7B (FDR).  *(c)* Vicuna-7B-V1.5 (FDR).

*(d)* Qwen2.5-3B (Power).  *(e)* Qwen2.5-7B (Power).  *(f)* Vicuna-7B-V1.5 (Power).

*Figure 13.* Test-time empirical selection-conditioned error rate (mean±std) and power (mean) on the TriviaQA dataset with LLM-as-a-Judge for correctness evaluation.

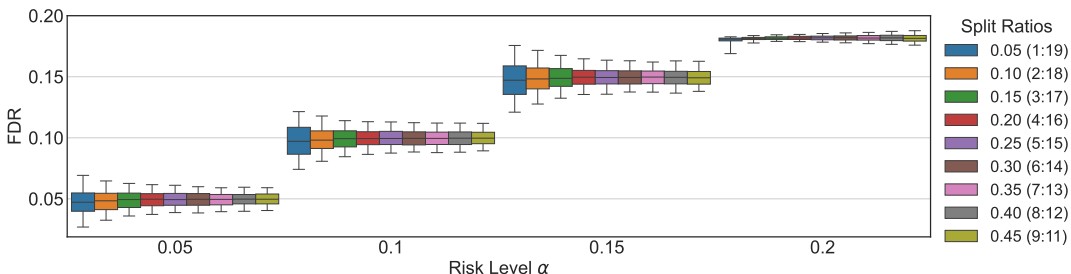

*Figure 14.* Empirical selection-conditioned error control across various calibration-test split ratios on CommonsenseQA with the OpenChat-3.5 model.

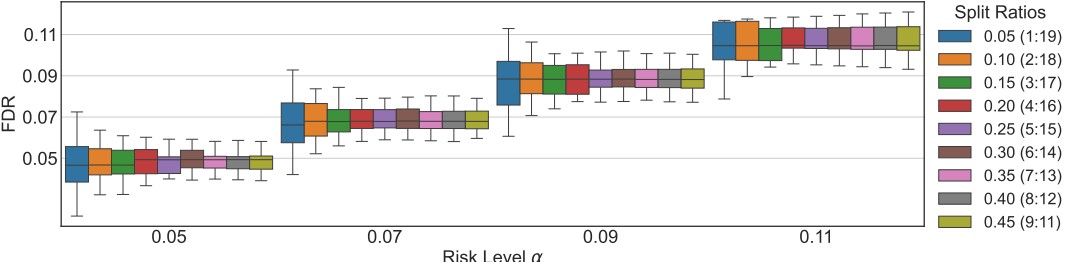

*Figure 15.* Empirical selection-conditioned error control across various calibration-test split ratios on TriviaQA with the LLaMA-3.1-8B model.

**Evaluation with LLM-as-a-Judge for Correctness Assessment.** Figure 13 reports the test-time selection-conditioned error rate and power on TriviaQA when correctness is evaluated using an LLM-as-a-Judge instead of exact string matching or similarity-based metrics. This setting introduces an additional layer of uncertainty, as the correctness labels themselves are noisy and may vary across prompts or judging criteria. Despite this increased label noise, `LEC` consistently maintains valid risk control across all evaluated models, with the empirical test-time selection-conditioned error rate closely tracking the target risk level and remaining below the theoretical upper bound. More importantly, `LEC` continues to achieve strictly higher power than both `UCB-HFD` and `UCB-CLP` across all models. The gap is especially pronounced for medium-sized models such as Qwen2.5-7B, where UCB-based methods suffer a sharp drop in power at low risk levels, while `LEC` is able to retain a substantial fraction of correct predictions. This behavior is consistent with the design of `LEC`: by enforcing a finite-sample linear constraint on the aggregate system behavior, `LEC` avoids overreacting to spurious or judge-induced

errors that disproportionately inflate confidence bounds in UCB-style calibration. Overall, these results demonstrate that `LEC` remains robust under noisy and subjective correctness evaluation schemes. This robustness is particularly important for open-ended question answering tasks, where exact correctness is often ill-defined and LLM-as-a-Judge–style evaluation is increasingly adopted in practice.

**Effect of Calibration–Test Split Ratios.** Figures 14 and 15 study the effect of different calibration–test split ratios. Across both CommonsenseQA and TriviaQA, LEC maintains valid risk control even when the calibration set is extremely small (e.g., split ratio 0.05, corresponding to only 500 calibration samples and 9500 test samples on CommonsenseQA). At the same time, these results clearly reflect the marginal nature of the theoretical guarantee: larger calibration sets lead to tighter concentration and reduced variance in test-time selection-conditioned error rate. As expected, increasing the calibration proportion reduces the standard deviation of the realized selection-conditioned error rate, reinforcing the practical benefit of allocating more data to calibration when possible.

**Evaluations on VQA Benchmarks.** We further evaluate the proposed `LEC` framework on two multimodal question answering benchmarks, open-ended MM-Vet v2 and closed-ended ScienceQA, using four representative LVLMs. As shown in Figures 16 and 17, `LEC` consistently achieves valid test-time risk control across all LVLMs and VQA datasets, while providing noticeably higher power compared to both UCB-based baselines. These results demonstrate that the benefits of `LEC` are not limited to language-only settings, but extend naturally to multimodal QA. Additionally, as demonstrated in Figures 18 and 19, `LEC` consistently maintains marginal guarantees across a wide range of calibration-test split ratios.

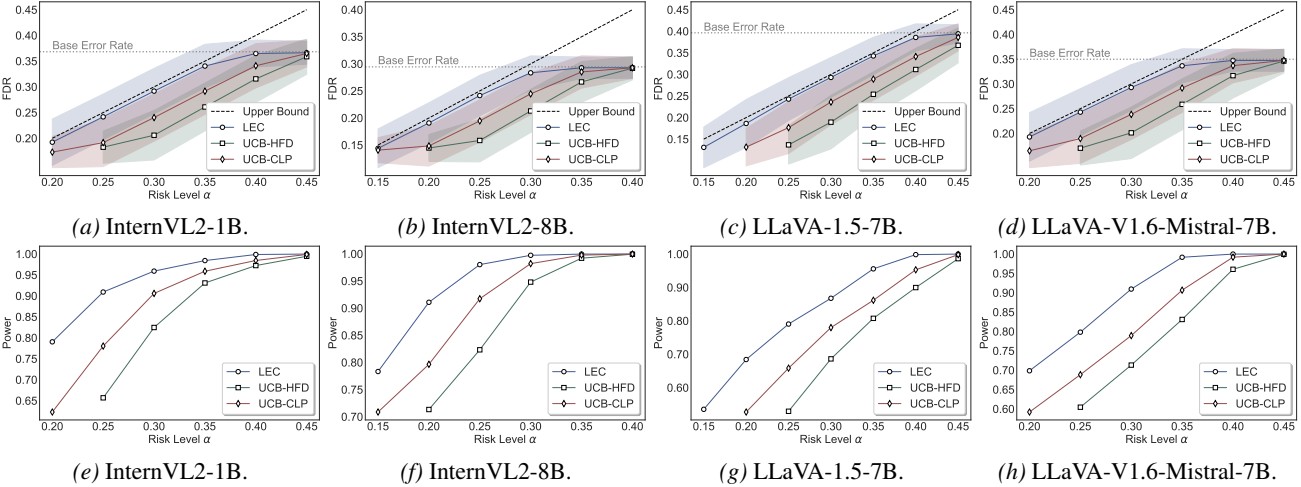

*(a)* InternVL2-1B.   *(b)* InternVL2-8B.   *(c)* LLaVA-1.5-7B.   *(d)* LLaVA-V1.6-Mistral-7B.

*(e)* InternVL2-1B.   *(f)* InternVL2-8B.   *(g)* LLaVA-1.5-7B.   *(h)* LLaVA-V1.6-Mistral-7B.

*Figure 16.* Test-time empirical selection-conditioned error rate (mean±std) and power (mean) on the MM-Vet v2 dataset.

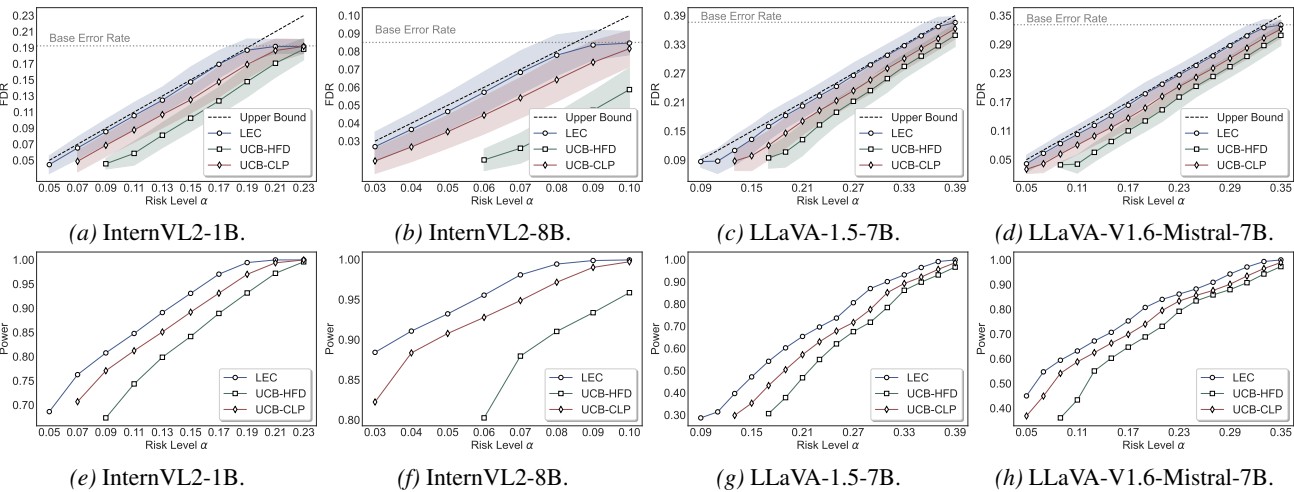

*(a)* InternVL2-1B.   *(b)* InternVL2-8B.   *(c)* LLaVA-1.5-7B.   *(d)* LLaVA-V1.6-Mistral-7B.

*(e)* InternVL2-1B.   *(f)* InternVL2-8B.   *(g)* LLaVA-1.5-7B.   *(h)* LLaVA-V1.6-Mistral-7B.

*Figure 17.* Test-time empirical selection-conditioned error rate (mean±std) and power (mean) on the ScienceQA dataset.

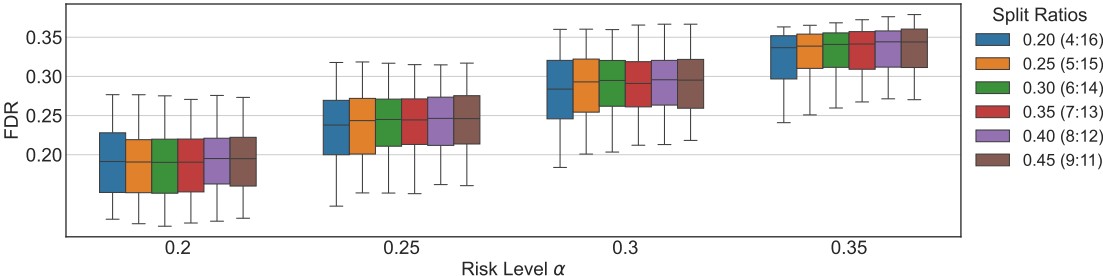

*Figure 18.* Empirical selection-conditioned error control across various split ratios on MM-Vet v2 with LLaVA-V1.6-Mistral-7B.

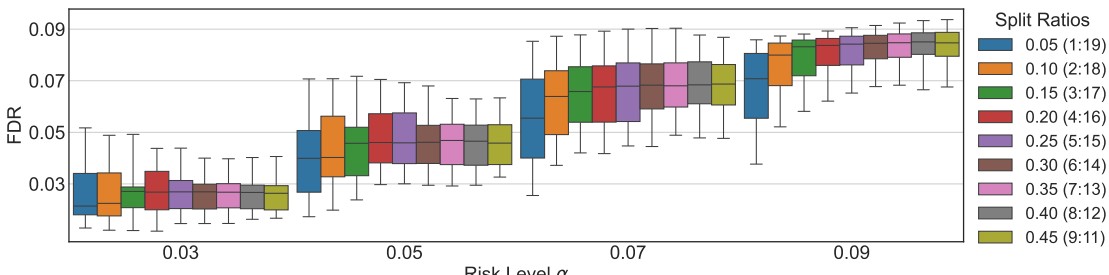

*Figure 19.* Empirical selection-conditioned error control across various split ratios on ScienceQA with InternVL2-8B.

**Two-Model Routing on TriviaQA.** Figure 20 reports the test-time selection-conditioned error rate of two-model routing systems on TriviaQA under different target risk levels. Across all model pairs, `LEC-Routing` consistently achieves tight risk control, operating close to the target risk level while remaining below the theoretical upper bound. In contrast, both `UCB-HFD-Routing` and `UCB-CLP-Routing` exhibit more conservative behavior, with realized selection-conditioned error rates staying noticeably below the target across most risk levels. Crucially, `LEC` calibrating independently for each model, without joint threshold calibration, tends to violate the target risk level or behave inconsistently across model pairs, since the resulting selection and error indicators no longer satisfy the finite-sample sufficient condition at the system level.

**Accepted Correct Samples under Two-Model Routing.** Table 5 further reports the allocation of accepted samples and the number of accepted correct predictions under two-model routing on the CommonsenseQA dataset. Compared to using either model alone, `LEC-Routing` consistently increases the number of accepted correct samples across model pairs and risk levels, while maintaining valid system-level risk control, as demonstrated in Figure 21. At $\alpha = 0.05$, for example, routing Qwen2.5-7B with LLaMA-3.1-70B under `LEC-Routing` increases the total acceptance rate from 50.69% (Qwen2.5-7B alone) and 55.57% (LLaMA-3.1-70B alone) to 57.09%, resulting in a higher number of accepted correct samples. Similar trends are observed at $\alpha = 0.10$ and for the Qwen2.5-14B pairing, where routing yields both higher coverage and more correct acceptances than either individual model. Compared to UCB-based routing methods, `LEC-Routing` achieves a more favorable balance between allocation efficiency and correctness, reflecting its tighter feasible region under finite-sample guarantees. Overall, these results suggest that joint calibration under the `LEC` framework can effectively leverage complementary strengths of multiple models to retain more correct predictions than single-model deployment, without sacrificing statistical reliability.

Beyond the total number of accepted correct samples, Table 5 highlights an important practical advantage of `LEC-Routing`: it tends to allocate a larger fraction of accepted samples to the cheaper primary model and invokes the more expensive secondary model only when necessary, while still improving overall correctness under the same risk budget. For instance, under the pair Qwen2.5-7B (primary) $\rightarrow$ LLaMA-3.1-70B (secondary) at $\alpha = 0.05$, `LEC-Routing` accepts 57.09% of test samples in total, where 43.77% are handled by the primary model and only 13.32% are delegated to the 70B model, yielding 2700 accepted correct samples. In contrast, `UCB-HFD-Routing` accepts substantially fewer samples overall (41.62%) and produces fewer correct acceptances (2016), despite still routing 18.44% of samples to the expensive model. `UCB-CLP-Routing` attains a similar total acceptance rate to `LEC-Routing` (56.95% vs. 57.09%) and comparable correct acceptances (2703 vs. 2700), but it requires routing a larger fraction to the 70B model (17.14% vs. 13.32%), implying higher inference cost for essentially the same utility.

Overall, these results suggest that `LEC-Routing` is not only statistically reliable but also operationally appealing: under

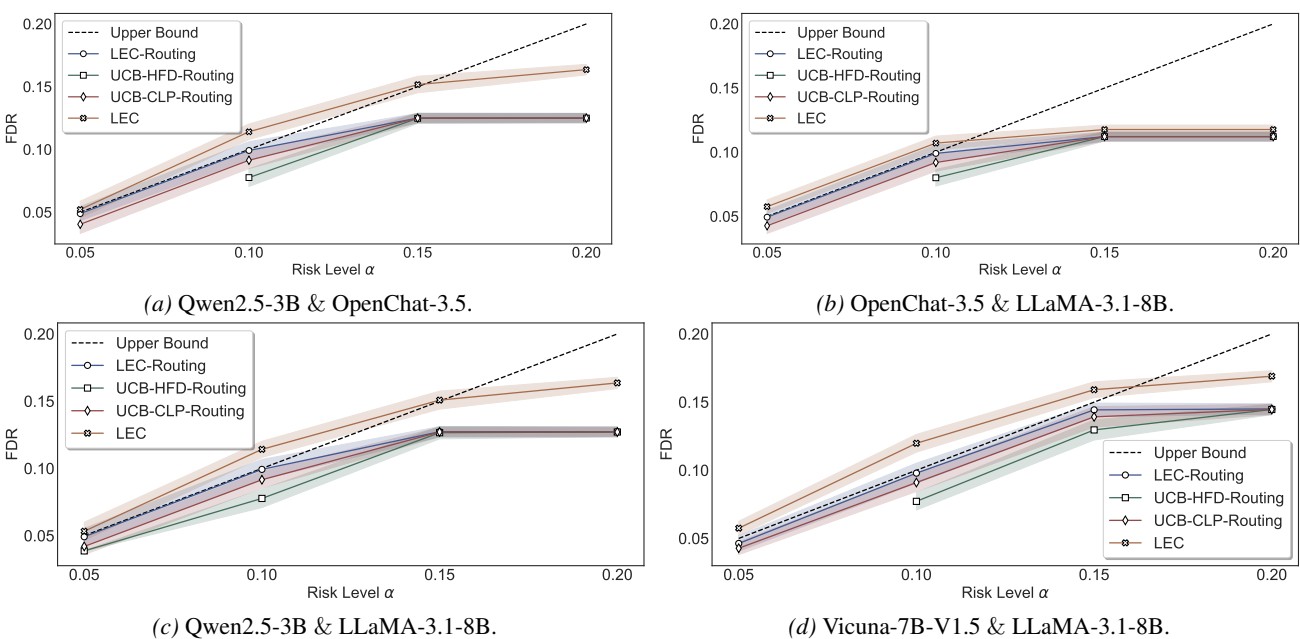

*Figure 20.* Test-time empirical system-level selection-conditioned error rate of two-model routing systems on TriviaQA (mean±std).

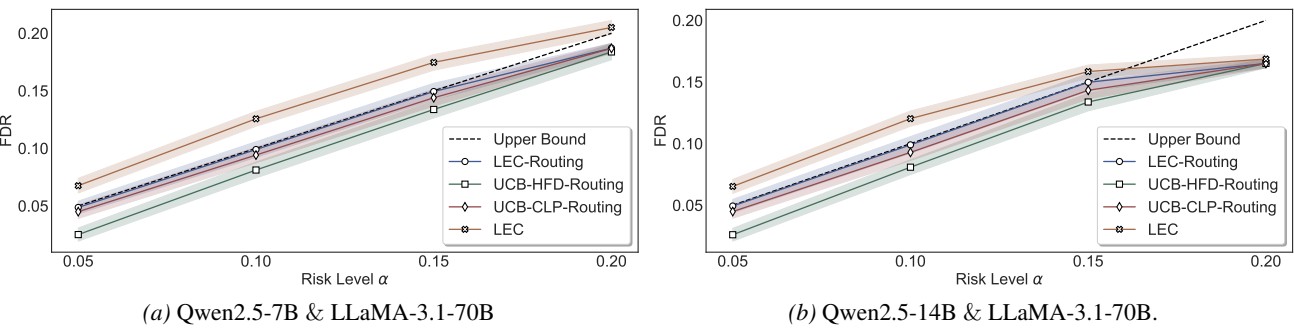

*Figure 21.* Test-time system-level selection-conditioned error rate of two-model routing systems on CommonsenseQA (mean±std).

finite-sample risk control, it can exploit the cheap model for the majority of accepted predictions and reserve the expensive model for genuinely uncertain cases, leading to a more favorable accuracy-cost trade-off in real deployments.

*Table 5.* Allocation of accepted samples (%) and accepted correct samples of two-model routing on the CommonsenseQA (mean).

| LLMs | Methods | $\alpha = 0.05$ | | | | $\alpha = 0.10$ | | | |
|------|---------|--------|--------|-------|------|--------|--------|-------|------|
| | | Prop. 1 | Prop. 2 | Total | Corr. | Prop. 1 | Prop. 2 | Total | Corr. |
| Qwen2.5-7B | LEC | 50.69 | - | 50.69 | 2393 | 68.81 | - | 68.81 | 3078 |
| LLaMA-3.1-70B | LEC | - | 55.57 | 55.57 | 2624 | - | 74.39 | 74.39 | 3329 |
| Qwen2.5-7B & LLaMA-3.1-70B | LEC-Routing | 43.77 | 13.32 | **57.09** | **2700** | 59.22 | 18.24 | **77.46** | **3469** |
| Qwen2.5-7B & LLaMA-3.1-70B | UCB-HFD-Routing | 23.18 | 18.44 | 41.62 | 2016 | 53.37 | 18.96 | 72.33 | 3303 |
| Qwen2.5-7B & LLaMA-3.1-70B | UCB-CLP-Routing | 39.81 | 17.14 | 56.95 | 2703 | 57.97 | 18.04 | 76.01 | 3422 |
| Qwen2.5-14B | LEC | 61.19 | - | 61.19 | 2789 | 81.60 | - | 81.60 | 3524 |
| LLaMA-3.1-70B | LEC | - | 57.34 | 57.34 | 2614 | - | 76.46 | 76.46 | 3302 |
| Qwen2.5-14B & LLaMA-3.1-70B | LEC-Routing | 55.91 | 7.58 | **63.49** | **2895** | 77.77 | 6.19 | **83.96** | **3629** |
| Qwen2.5-14B & LLaMA-3.1-70B | UCB-HFD-Routing | 32.80 | 15.53 | 48.33 | 2257 | 73.19 | 4.57 | 77.76 | 3428 |
| Qwen2.5-14B & LLaMA-3.1-70B | UCB-CLP-Routing | 53.66 | 7.31 | 60.97 | 2793 | 75.83 | 6.48 | 82.31 | 3581 |

