# OpenReview forum: "LEC: Linear Expectation Constraints for Selection-Conditioned Risk Control in Selective Prediction and Routing Systems"
_ICML.cc/2026/Conference — ICML 2026 regular_

### Official Review · Reviewer_mPTs · 2026-03-10

**Soundness:** 3
**Presentation:** 3
**Significance:** 3
**Originality:** 3
**Overall Recommendation:** 4
**Confidence:** 3

**Summary:**

This paper considers selective prediction for foundation models under FDR control. The main idea is to formulate FDR control through a linear expectation constraint over selection and error indicators, and then use a held-out calibration set to choose uncertainty thresholds through a finite-sample sufficient condition. Assuming exchangeability of data, the paper provides marginal guarantees for both the single-model setting and a two-model routing system. Experiments on QA and VQA tasks across multiple LLMs/LVLMs and uncertainty estimators, show that LEC achieves valid test-time FDR control while generally attaining higher power than UCB-based baselines. In the routing setting, it also appears to route queries more efficiently, allowing a larger fraction of accepted samples to remain with the cheaper model without losing the reliability guarantee.

**Compliance With Llm Reviewing Policy:**

Affirmed.

**Final Justification:**

My concerns have been largely addressed, and I will maintain my current score.

**Key Questions For Authors:**

1. Since the theoretical guarantees are marginal over calibration and test randomness, could the authors clarify how this guarantee should be interpreted in deployment, where the calibrated threshold is fixed and reused on many future queries?

2. How is the optimization over the routing threshold pair implemented in practice, and what is its computational cost?

3. The open-ended evaluation depends on the chosen admission function. How robust are the empirical conclusions to this choice?

**Limitations:**

No. It would help to talk more about the exchangeability assumption and possible distribution shift, the fact that the guarantees are marginal rather than conditional on the realized calibration set and the dependence of the open-ended evaluation on the chosen admission function.

**Strengths And Weaknesses:**

**Strengths**

- The paper is built on a clean idea, and the formulation is easy to follow.

- The problem is practically relevant, especially for routing and selective prediction in foundation model systems.

- The experiments are generally solid and support the main empirical claims.


**Weaknesses**

- The guarantees are marginal over calibration and test randomness, rather than conditional on the realized calibration set. As a result, the paper could do more to clarify how this guarantee can be interpreted in deployment, where the calibrated threshold is fixed and then used on many future queries.

- The empirical comparison is somewhat narrow, and the open-ended evaluation depends on the chosen admission function.

---

> ### Author Rebuttal · Authors · 2026-03-26
>
> Thank you for these thoughtful comments.
> > W1
>
> We agree that the deployment interpretation of our guarantee should be explained more clearly.
>
> Our guarantee is **marginal over both calibration and future test randomness**, rather than conditional on a realized calibration set. Therefore, we do **not** claim an exact conditional guarantee for a single fixed calibrated threshold after one particular calibration split. Instead, the correct interpretation is the standard one in split conformal prediction: if the calibration set and future deployment queries are drawn from the same exchangeable/data-generating pipeline, then across repeated deployments the **long-run selective risk** is controlled at the target level. This is also the standard statistical target in conformal prediction, since **exact distribution-free conditional guarantees are generally unattainable without stronger assumptions** [1].
>
> In deployment, this means that the procedure is statistically valid at the **population level** under the assumed sampling model. This aligns with standard split conformal methods: the guarantee is meaningful for repeated use under the same deployment conditions, rather than as a deterministic guarantee conditional on one fixed calibration set.
> > W2
>
> We agree that robustness to the chosen admission function is important in open-ended evaluation.
>
> At the same time, we would like to clarify that our empirical study is not restricted to a single admission rule. **In the appendix, we already evaluate open-ended tasks using three representative admission functions: sentence similarity, bi-entailment, and LLM-as-judge**. These cover the main paradigms currently used for open-ended correctness assessment. Our main empirical conclusions are consistent across these choices.
>
> More importantly, **the proposed guarantee is agnostic to the specific admission function**. The admission rule defines the operational notion of correctness, while our method provides statistical control with respect to that chosen notion. As long as the same admission function is used consistently in both calibration and test, under the same exchangeability assumptions, the statistical validity of our method is preserved. In this sense, changing the admission function changes the target notion of correctness, but does not invalidate the risk-control guarantee itself.
>
> We appreciate the reviewer’s suggestion and will revise the paper to emphasize more clearly in the main text:
> - Robustness across multiple admission functions is already included in the appendix.
> - Our theoretical framework is admission-function-agnostic as long as the evaluation rule is fixed consistently across calibration and deployment.
>
> > Q1
>
> Let $D_{cal}$ denote the realized calibration set and let $\hat \lambda$ be the calibrated threshold. Once deployed, this threshold is fixed and reused on future queries. The relevant deployment quantity is therefore the realized selective risk $R_{dep} (D_{cal}) := Pr(err=1 | S_{\hat \lambda} (X)=1, D_{cal})$, where the probability is over future test queries, conditional on the realized calibration set.
>
> Our theory does not claim that for every realized $D_{cal}$, $R_{dep} (D_{cal}) \le \alpha$. Rather, it establishes the marginal guarantee $\mathbb{E}[R_{dep} (D_{cal})]\le \alpha$, i.e., the deployment risk is controlled **on average over calibration randomness**. In practice, once $D_{cal}$ is fixed, the empirical selective risk over many future queries concentrates around $R_{dep} (D_{cal})$.
> > Q2
>
> In practice, the routing threshold pair is optimized **offline** on the calibration set by searching over candidate threshold pairs induced by the observed uncertainty scores and selecting the best feasible pair.
>
> To quantify its computational cost, we measure the wall-clock time of a single offline joint-threshold calibration run on CommonsenseQA using 10,000 samples with split ratio 0.5 (i.e., 5,000 calibration / 5,000 test) at target risk level $\alpha=0.05$. Under this setting, the average calibration time is 1.967 seconds.
>
> Importantly, this cost is incurred only **once during calibration**; after that, the threshold pair is fixed, and test-time routing requires only uncertainty computation plus a simple threshold decision. Thus, this is an **offline calibration cost**, not a deployment-time latency bottleneck. Cumulative bookkeeping can further reduce calibration cost, but threshold-search optimization is not the main focus of this paper.
> > Q3
>
> Thank you for this important question. In the appendix, we already evaluate sentence similarity, bi-entailment, and LLM-as-judge, and the qualitative conclusions remain consistent. Moreover, our framework is admission-function-agnostic. As long as the same admission function is used consistently in calibration and test under exchangeability, the validity guarantee is preserved.
>
> [1] A Gentle Introduction to Conformal Prediction and Distribution-Free Uncertainty Quantification.

---

> > ### Author Rebuttal · Reviewer_mPTs · 2026-04-03
> >
> > Thank you for the detailed response. My main concerns have been adequately addressed. In particular, the authors have clarified the deployment interpretation of the marginal guarantee, explained the practical optimization cost of the routing threshold pair, and addressed my question about robustness to the admission function. I have no further questions.

---

> > > ### Author Response · Authors · 2026-04-03
> > >
> > > Thank you very much for your thoughtful follow-up and for your encouraging feedback. We are very glad that our rebuttal has adequately addressed your concerns. We sincerely appreciate your careful review and insightful suggestions, which have helped us improve both the clarity and positioning of the paper.

---

### Official Review · Reviewer_9sAd · 2026-03-12

**Soundness:** 2
**Presentation:** 3
**Significance:** 2
**Originality:** 3
**Overall Recommendation:** 4
**Confidence:** 4

**Summary:**

This paper introduces a principled method called LEC for the selective prediction problem. LEC approaches the problem from the perspective of false discovery rate (FDR) control, ensuring that the proportion of errors among all accepted predictions stays below a specified risk level. Experimental results show that the proposed LEC method controls the FDR and achieves higher power in both single-model and two-model settings.

**Compliance With Llm Reviewing Policy:**

Affirmed.

**Final Justification:**

The issue in Weakness-1 stems from a misuse of concepts and can be corrected without affecting the subsequent content. Weakness-2 still remains; however, given that the major issue in Weakness-1 can be fixed, I think the paper merits a “weak accept” rating.

**Key Questions For Authors:**

1. Are the singe-model and two-model settings commonly used in selective prediction community?
2. What does "Base Error Rate" refer to in the figures?

Other questions see Weaknesses.

**Limitations:**

The authors discusse some limitations, and my suggestions for improvement can be seen in Weaknesses.

**Strengths And Weaknesses:**

Strengths:
1. The LEC method guarantees FDR control, ensuring that among all accepted predictions, the proportion of errors does not exceed a target risk level.
2. Extensive experimental results demonstrate that LEC achieves higher statistical power than UCB-HFD and UCB-CLP in both single-model and two-model settings.

Weaknesses:
1. In Equation (3), the definition of false discovery rate (FDR) is misused. In the statistical literature [1, 2], FDR is formally defined as the expectation of the false discovery proportion (FDP), where in the context of this paper, $\text{FDP} = Z^a(\lambda^a) / S^a(\lambda^a)$. Accordingly, $\text{FDR} = \mathbb{E}[\text{FDP}] = \mathbb{E}[Z^a(\lambda^a) / S^a(\lambda^a)]$, which is not equal to $\mathbb{E}[Z^a(\lambda^a)] / \mathbb{E}[S^a(\lambda^a)]$. This misuse undermines the subsequent derivation about linear expextation.


2. The paper claims that LEC achieves higher statistical power than existing FDR-based methods such as UCB-HFD and UCB-CLP. However, this conclusion is supported only through experimental validation, without a theoretical derivation. This limits the contribution of the paper.

Reference:

[1]. Benjamini, Yoav, and Yosef Hochberg. "Controlling the false discovery rate: a practical and powerful approach to multiple testing." Journal of the Royal statistical society: series B (Methodological) 57.1 (1995): 289-300.

[2]. Jin, Ying, and Emmanuel J. Candès. "Selection by prediction with conformal p-values." Journal of Machine Learning Research 24.244 (2023): 1-41.

---

> ### Author Rebuttal · Authors · 2026-03-26
>
> Thank you for these thoughtful comments.
> > Response to Weakness 1
>
> We appreciate the reviewer for pointing out this, and we agree that Eq. (3) conflates the classical multiple-testing notion of FDR with the quantity actually targeted in our paper. Under the classical definition, $FDR=\mathbb{E}[FDP]$, which is in general not equal to $\mathbb{E}[Z]/\mathbb{E}[S]$. Our intended target is instead the **selection-conditioned error rate (equivalently, marginal FDR)** for a randomly drawn test example. For binary $S$ and $Z=S\cdot err$, we have $\mathbb{E}[Z]=Pr(S=1,err=1)$, $\mathbb{E}[S]=Pr(S=1)$, so $\mathbb{E}[Z]/\mathbb{E}[S]=Pr(err=1|S=1)$.
>
> This is exactly the target stated in Eq. (1)-(2) and guaranteed in Theorems 3.1–3.2. Therefore, the issue is confined to the **definition/terminology in Eq. (3)**, rather than the validity of the subsequent guarantee. We will revise Eq. (3) accordingly, reserve “FDR” for the classical $\mathbb{E}[FDP]$ notion, and refer to our target more precisely as **marginal FDR or selection-conditioned error rate**.
> > Response to Weakness 2
>
> We agree that the current manuscript does not prove a universal theorem that LEC always has higher statistical power than UCB-based methods. We will soften this claim. However, LEC is theoretically less conservative from the **calibration rule** itself. For a threshold $\lambda$, let $k(\lambda)$ be the number of accepted calibration samples and $m(\lambda)$ the number of errors among them. LEC requires $m(\lambda) - \alpha k(\lambda) \le -1$, i.e., $m(\lambda) \le  \alpha k(\lambda) -1$.
>
> **By contrast**, Hoeffding-UCB requires $\frac{m(\lambda)}{k(\lambda)}+\sqrt{\frac{log(1/ \delta)}{2k(\lambda)}} \le \alpha$, equivalently $m(\lambda)\le \alpha k(\lambda) - \sqrt{\frac{k(\lambda) log(1/ \delta)}{2}}$, so its correction scales as $O(\sqrt{k(\lambda)})$, whereas LEC uses only an $O(1)$ correction. For exact Clopper–Pearson UCB, requiring the upper bound to be below $\alpha$ means $m(\lambda)$ must fall in the lower tail of a $Binomial(k(\lambda),\alpha)$ distribution, which asymptotically again requires a $\Theta (\sqrt{k(\lambda)})$ downward deviation below $\alpha k(\lambda)$. Thus, both UCB baselines admit a smaller feasible threshold region and are expected to be more conservative. We will revise the paper to state this more precisely: **LEC has a principled reason to be less conservative**, and this consistently translates into stronger empirical power in our experiments.
> > Response to Question 1
>
> - For the single-model setting, the setup directly matches the classical selective prediction framework: a predictor produces an output together with an uncertainty score, and the system either **accepts** the prediction or **abstains** based on a threshold. This is the canonical setting in selective prediction, where the central goal is to control the error among accepted predictions while maximizing coverage. In our paper, the single-model case plays exactly this role.
>
> - For the two-model setting, it is a very natural and practically important extension of the same idea: instead of abstaining after rejecting the first model, the system **routes** the sample to a second model. This preserves the core selective prediction principle (using uncertainty to decide whether the current prediction should be trusted) while replacing abstention with cascading.
>
> This two-model setting is well motivated in several applications. For example, in **cloud-edge collaboration**, a lightweight edge model handles easy cases locally, while uncertain cases are routed to a more capable cloud model. The key problem is still selective prediction: the system knows when the first model can be trusted and when escalation is necessary.
> > Response to Question 2
>
> By “Base Error Rate”, we mean the model’s overall error rate on the full test set when **no selective prediction mechanism is applied**, i.e., when the model is used on all test samples without rejection, abstention, or routing.
> Since all results are obtained over 500 random calibration/test splits, the “Base Error Rate” is the average full-test-set error rate across the 500 runs.
>
> We use this quantity as a reference point to show the benefit of selective prediction: whether the error among accepted predictions is reduced relative to applying the model indiscriminately to the entire test set.
> > Summary
>
> We sincerely thank the reviewer for these insightful comments, which help clarify both the theoretical positioning and practical implications of our work. We will refine the definition to distinguish the target guarantee and clarify the comparison with UCB baselines.
>
> Overall, our goal is to provide a practical and less conservative risk-control framework for selective prediction, enabling more effective utilization of accepted predictions under a specified error constraint. We believe this perspective complements existing UCB-style approaches and offers a useful alternative for real-world deployment.

---

> > ### Author Rebuttal · Reviewer_9sAd · 2026-04-04
> >
> > Thank the authors for their response. I hope the revisions will be incorporated into the new version. I will improve the rating to weak accept.

---

> > > ### Author Response · Authors · 2026-04-04
> > >
> > > Thank you very much for your thoughtful follow-up and for your positive update. We truly appreciate your careful reading and insightful feedback throughout the review process. We will make sure to incorporate the discussed revisions into the new version. Thank you again for your time and support.

---

### Official Review · Reviewer_uCSw · 2026-03-12

**Soundness:** 3
**Presentation:** 3
**Significance:** 3
**Originality:** 2
**Overall Recommendation:** 4
**Confidence:** 3

**Summary:**

This paper proposes a a framework for controlling the false discovery rate (FDR) among accepted predictions from foundation models (LEC). The main idea is to reformulate FDR control as a linear expectation constraint $E[Z - \alpha S] \leq 0$ over binary selection and error indicators, then derive a finite-sample sufficient condition via a "+1 correction" under exchangeability. The resulting calibrated threshold maximizes acceptance coverage while guaranteeing $FDR \leq \alpha$. The framework extends to two-model routing, where rejected inputs from a primary model are forwarded to a secondary model with joint threshold calibration preserving system-level FDR control. Experiments across QA and VQA benchmarks with 8 LLMs and 4 LVLMs show LEC achieves tighter FDR control and higher power (sample retention) than UCB-based baselines.

**Compliance With Llm Reviewing Policy:**

Affirmed.

**Final Justification:**

This paper proposes a well-motivated framework for FDR control in selective prediction from foundation models. The formulation is elegant, the evaluation is comprehensive, and the empirical results are good.

My concerns (the confounded guarantee comparison, missing computation costs, and lack of routing benefit analysis) were adequately addressed in the rebuttal. The exceedance rate data (Q1) and runtime figures (Q2) are useful additions, and the informal characterization of routing conditions (Q3) is reasonable given the paper's scope. I maintain my positive score (weak accept).

**Key Questions For Authors:**

1. What fraction of individual calibration splits (out of 500) result in test-time FDR exceeding $\alpha$?
2. What is the wall-clock time for the joint threshold calibration in the two-model routing setting? How does this scale with calibration set size?
3. Can you characterize, even informally, conditions on model pair properties (e.g., error correlation, accuracy differential) under which routing provides substantial benefit over the better single model?

**Limitations:**

Yes

**Strengths And Weaknesses:**

Strengths

- Well-motivated & clean formulation. Controlling the error rate among accepted predictions is arguably more useful for deployment than marginal coverage guarantees from conformal prediction, since users only interact with accepted outputs. The reformulation of FDR as $E[Z - \alpha S] \leq 0$ is elegant and yields a simple, easy-to-implement calibration procedure. The observation that the same linear decomposition applies at the system level for routing is nice. The results that independent per-model calibration violates system-level guarantees (Figures 6, 20) motivates joint calibration.
- Thorough experiments. The evaluation is comprehensive across multiple dimensions: 8 LLMs and 4 LVLMs, 4 benchmarks spanning closed-ended and open-ended QA/VQA, 6 uncertainty estimators, 3 alignment criteria, varying calibration-test split ratios, varying sampling sizes, and 500 random splits
- Strong empirical results. LEC uniformly outperforms UCB-based baselines, with larger gains at low risk levels. The routing results (Table 2, Table 5) show that joint calibration increases accepted correct samples beyond either model alone, with cost-efficient allocation favoring the cheaper primary model.
- Useful practical insights in the routing analysis. The allocation analysis (Figure 7, Figure 8, Table 5) goes beyond simply validating FDR control to show that LEC-Routing preferentially uses the cheaper primary model and routes to the expensive secondary model only when necessary. This cost-efficiency dimension is important for real deployments.

Weaknesses
- The comparison with UCB baselines is confounded by different guarantee types. LEC provides a marginal FDR guarantee (averaged over both calibration and test randomness), while UCB-based methods provide PAC-style guarantees that hold with high probability over the calibration set. LEC's tighter empirical FDR are partly an expected consequence of this weaker guarantee. The paper could present this more clearly by reporting the fraction of individual calibration splits (out of 500) where test-time FDR exceeds $\alpha$. This would clarify how the marginal guarantee behaves conditionally and make the comparison more fair.
- Computational cost. Algorithm 3 searches over all pairs of candidate thresholds, which is $O(n^2)$ pairs each requiring $O(n)$ evaluation. No computation times are reported. The multi-model extension in Appendix B scales exponentially in the number of models.
- No discussion of when routing is beneficial. The paper demonstrates empirically that routing can increase accepted correct samples, but provides no analysis of what model pair properties (accuracy gap, uncertainty correlation, complementary error patterns) determine whether routing yields substantial gains versus marginal ones. The paper acknowledges this as future work, but some discussion would be nice.

---

> ### Author Rebuttal · Authors · 2026-03-26
>
> Thank you for these insightful comments.
> > W1
>
> We agree that the guarantee notions are different and that this should be stated more clearly. At the same time, LEC and UCB-based baselines **share the same operational objective** in selective prediction: calibrating a threshold to control the error rate among accepted predictions.
>
> LEC  targets **marginal selective validity**, i.e., average control over calibration and test randomness, which is aligned with the standard split-conformal style notion of statistical validity [1]. UCB baselines follow the same split-calibrate-then-test paradigm and target the same selective risk, but use a high-probability worst-case calibration guarantee. They are therefore more conservative, **but they are still not deterministic conditional guarantees** for every split: failures can still occur.
>
> So we agree that the guarantee types are different, but we believe the comparison is still meaningful because the underlying selective prediction objective is the same. Our contribution is to obtain **tighter marginal risk control**, which is theoretically meaningful and practically useful because it keeps realized FDR closer to the target level and improves coverage under the same selective prediction goal.
> > W2
>
> We agree that the current manuscript does not discuss computation time clearly enough. **The key point, however, is that the threshold search happens only in the offline calibration stage and does not affect deployment-time usage.**
>
> Thus, the concern here is mainly about one-time offline calibration cost, whereas the online cost is only uncertainty computation plus a fixed accept or routing decision. This separation between offline calibration and fixed-threshold test-time routing is standard in closely related cascade-style frameworks, such as *Trust or Escalate* [2].
>
> In practice, the search can be accelerated substantially through cumulative bookkeeping or vectorized evaluation. We agree that the **naive multi-model extension** grows quickly with the number of models, and we will clarify that our main empirical focus is on the single-model and two-model settings, where calibration remains practical.
> > W3
>
> We agree that routing is **not universally beneficial**. Our goal in the two-model setting is not to claim that A+B always outperforms always using the stronger model B.
>
> If cost is irrelevant and the only objective is maximum accuracy, always using B may indeed be preferable. Rather, the motivation for routing is to achieve a better **cost–accuracy trade-off under a user-specified risk level**: once the target risk is satisfied, maximizing the safe use of the cheaper model A and reducing calls to the expensive model B is already a meaningful deployment benefit.
>
> We agree that systematically characterizing these conditions would strengthen the paper, but we view this as beyond the current scope. **Our current claim is intentionally modest: routing has the potential to enlarge the effective set of correct predictions and improve cost-efficiency, rather than universally dominating always using B.**
> > Q1
>
> On CommonsenseQA using LLaMA-3.1-8B with 5,000 calibration + 5,000 test samples, target risk $\alpha=0.05$ over 500 random splits:
> | Method | Exceedance | FDR | Accept Correct (%) |
> |-|-|-|-|
> |LEC|235/500|0.0494±0.0078|43.41%|
> |UCB-CLP|57/500|0.0408±0.0078|39.02%|
> |UCB-HFD|nan|nan|nan|
>
> LEC targets **marginal guarantee**: it is expected that some splits exceed $\alpha$, while the mean FDR remains below $\alpha$ (**validity is supported by Figure 11 in [1]**). UCB-CLP has fewer exceedances, but this also leads to lower coverage. UCB-HFD is more conservative and fails to find a feasible threshold during calibration.
> > Q2
>
> We report the wall-clock time of a single offline joint-threshold calibration run in the two-model routing setting on CommonsenseQA, using LLaMA-3.1-8B + LLaMA-3.1-72B:
> | Calibration-Test Num | Runtime (sec)|
> |-|-|
> |5000-5000|1.967±0.035|
> |2500-7500|0.774±0.022|
> |1000-9000|0.258±0.020|
>
> Since the joint threshold search is performed only on the calibration set, its runtime scales primarily with the calibration size. Importantly, this cost is incurred **only once offline**,  and it does not affect deployment-time latency, which is also standard in related cascade-style selective evaluation methods.
> > Q3
>
> Informally, routing is most beneficial when: (i) model B is substantially stronger but also more expensive than model A; (ii) model A's uncertainty is informative enough to identify cases that should be escalated; and (iii) model B  has a clear advantage on the routed subset (i.e., some degree of error complementarity). If A and B make highly correlated errors, or if B's gain on the uncertain subset is small, then routing over the better single model is naturally limited.
>
> [1] A Gentle Introduction to Conformal Prediction and  Distribution-Free Uncertainty Quantification.
>
> [2] Trust or Escalate: LLM Judges with Provable Guarantees for Human Agreement.

---

> > ### Author Rebuttal · Reviewer_uCSw · 2026-04-03
> >
> > Thanks for the rebuttal and additional results. Your clarifications addressed my questions, and I have no further questions. I will keep my current score.

---

> > > ### Author Response · Authors · 2026-04-04
> > >
> > > Thank you very much for your careful review and for taking the time to follow up on our rebuttal. We truly appreciate your thoughtful feedback throughout the process.
> > >
> > > We are glad that our clarifications and additional results have addressed your questions and helped improve the presentation and clarity of the work. We sincerely thank you again for your insightful comments and support.

---

### Decision · Program_Chairs · 2026-04-30

**Decision:**

Accept (regular)

**Comment:**

The reviewers found the paper to be a well-motivated and practically relevant contribution, highlighting the clean formulation of LEC for selective prediction and routing, the strong empirical evaluation across QA/VQA settings, and the consistent gains in retention/power relative to UCB-style baselines. The main weaknesses concerned the distinction between marginal and conditional guarantees, the fairness of comparing LEC to baselines with different guarantee types, the initial imprecision in the paper’s use of FDR terminology, and limited discussion of calibration cost, deployment interpretation, and when routing is most beneficial. In rebuttal, the authors directly addressed these concerns by acknowledging and correcting the terminology issue, clarifying that the guarantee is a marginal split-conformal style guarantee, providing additional exceedance-rate and runtime results, and giving a more careful characterization of the settings in which routing yields useful cost–accuracy tradeoffs. Overall, the concerns were satisfactorily addressed and the final reviewer consensus is positive. This appears to be a useful contribution with clear practical relevance, while some limitations remain in theoretical positioning and scope.